# Oxytocin enhances observational fear in mice

Marc T. Pisansky [1,2], Leah R. Hanson[3], Irving I. Gottesman [4] & Jonathan C. Gewirtz[2,4]

Empathy is fundamental to human relations, but its neural substrates remain largely unknown. Here we characterize the involvement of oxytocin in the capacity of mice to display emotional state-matching, an empathy-like behavior. When exposed to a familiar conspecific demonstrator in distress, an observer mouse becomes fearful, as indicated by a tendency to freeze and subsequent efforts to escape. Both intranasal oxytocin administration and chemogenetic stimulation of oxytocin neurons render males sensitive to the distress of an unfamiliar mouse. Acute intranasal oxytocin penetrates the brain and enhances cellular activity within the anterior cingulate cortex, whereas chronic administration produces long-term facilitation of observational fear and downregulates oxytocin receptor expression in the amygdala. None of these manipulations affect fear acquired as a result of direct experience with the stressor. Hence, these results implicate oxytocin in observational fear in mice (rather than fear itself) and provide new avenues for examining the neural substrates of empathy.

[1] Graduate Program in Neuroscience, University of Minnesota—Twin Cities, Jackson Hall 6-145, 321 Church St SE, Minneapolis, MN 55455, USA. [2] Department of Neuroscience, University of Minnesota—Twin Cities, WMBB 4-280, 2102 6th street SE, Minneapolis, MN 55455, USA. [3] Neuroscience Research, HealthPartners Research Foundation, 295 Phalen Boulevard, St. Paul, MN 55130, USA. [4] Department of Psychology, University of Minnesota—Twin Cities, Elliott Hall N246, 75 E River Road, Minneapolis, MN 55455, USA. Irving I. Gottesman is Deceased. Correspondence and requests for materials should be addressed to J.C.G. (email: jgewirtz@umn.edu)

Empathy refers to the ability of an individual to infer and respond to the emotional states of others[1]. This facility is vital for normal social behavior and impaired across several psychiatric diseases[2–4]. Long acknowledged in primates, recent findings have begun to identify capacities for social cognition processes in rodents as well. For example, several studies have now documented socially transmitted, or observational, fear in rodents observing the distress of a conspecific[5–10]. These studies highlight the conserved nature of emotional state-matching. Nevertheless, the underlying neurobiological substrates of this component of empathy are largely unknown.

One molecular substrate of empathy is oxytocin, a neuropeptide classically implicated in parturition and mother-infant care. Intranasal administration of oxytocin to human subjects enhances emotional recognition[11] and empathy[12] by modulating amygdala activity[12] and amygdala-cortical connectivity[13,14]. Oxytocin is conserved in rodents where it has been suggested to moderate fear[15–18] and anxiety[19–21], as well as various social behaviors. For instance, oxytocin signaling plays a role in social approach[22,23], recognition[24], buffering[25,26], reward[27], and learning[28]. A recent study in prairie voles has demonstrated the importance of oxytocin in consolation of a stressed conspecific[29], an empathy-like form of behavior. However, it remains unclear whether oxytocin plays a similarly important role in empathic behavior in mice.

In this study, we investigate the effects of oxytocin on mice observing the distress of a conspecific using a novel set of socially transmitted fear paradigms. We find that mice exhibit emotional state-matching (fear contagion or observational fear) in response to the distress of a familiar conspecific, expressed as initial freezing and subsequent escape behavior. Conversely, unfamiliar male mice exhibit lower levels of these social fear behaviors. Freezing-specific behavior is rescued through intranasal oxytocin or chemogenetic activation of hypothalamic oxytocinergic neurons. Acute intranasal oxytocin successfully localizes to multiple brain regions, including the anterior cingulate cortex (ACC), amygdala, and hypothalamus. Administered chronically, oxytocin produces similar enhancements in emotional state-matching measured several days after cessation of drug treatment and downregulates oxytocin receptor expression in the central lateral nucleus of the amygdala (CeL). We also find a relationship between litter size and social fear behavior, implicating early-life social experience in the maturation of empathy. Importantly, whereas oxytocin enhances socially acquired fear, it consistently has no effect on fear conditioned to nonsocial (i.e., auditory and contextual) cues. Hence, rather than moderating levels of fear per se, the oxytocin system appears to be a critical substrate for socially transmitted fear.

## Results

**Individual differences in social fear.** To investigate empathic behavior in mice, we have devised a social fear paradigm in which Pavlovian fear conditioning of a demonstrator mouse produces freezing in a same-sex observer mouse[30] (Fig. 1a, top; Supplementary Movie 1). That our conditioning procedure effectively produced fear in the demonstrator mice was confirmed in a fear-potentiated startle test conducted the day after conditioning (Supplementary Fig. 1). The load cell transducer system used to measure startle was adapted to quantify decreases in force-generated locomotor activity of observers across several intra-trial measurements. Quantifying measurements below threshold produced a reliable proxy for freezing (Supplementary Fig. 2). Non-specific freezing by observer mice were measured during control experiments conducted 2 days prior to conditioning (Fig. 1a, bottom). In the first control experiment electrical current was passed through an empty demonstrator cage (noDem); in the second, the demonstrator was present but not conditioned (noShock).

We first investigated the effects of sex and familiarity on socially transmitted fear behavior. While social fear behaviors in mice are known to be influenced by familiarity (i.e., familiarity bias)[5,31], sex-specific effects have been less studied (see ref. [9]). In female mice, we discerned no effect of familiarity—that is, sibling (henceforth, familiar) and non-sibling/non-cagemate (henceforth, unfamiliar) observers exhibited high and comparable levels of freezing during demonstrator conditioning (Fig. 1b, right). We also detected no difference in non-specific freezing behavior of females during control experiments (Supplementary Fig. 3). In contrast, unfamiliar male observers exhibited a significant deficit in freezing compared to familiar observers— both when quantified across conditioning trials (Fig. 1b, left) and when averaged (Fig. 1b, right). In a subset of familiar and unfamiliar males, we found that the degree of freezing of the observer correlated with freezing of the demonstrator when the two animals were familiar, but not when they were unfamiliar (Supplementary Fig. 4).

This familiarity effect in males was not attributable to the duration or characteristics of individual vocalizations recorded from demonstrator mice. However, compared to males, we noted a significantly elevated duration of vocalizations from females, implying either enhanced nociceptive sensitivity[32] or heightened communication of distress (Supplementary Fig. 5). Vocalizations contribute to socially acquired fear in rats[8]; therefore we also analyzed the relationship between demonstrator vocalizations and observer freezing across conditioning trials by individual pairs of mice. Familiar males showed a significantly higher average correlation compared to unfamiliar males (Fig. 1c). This finding suggests that vocalizations contribute to social fear exclusively between familiar conspecifics, even though there were no discernable qualitative differences in their expression between familiar and unfamiliar mice.

Early-life environment contributes strongly to the development of emotional processes and social behavior[33]. Such an influence is suggested in the current data in that familiar observers raised in smaller litters (2–6 pups) exhibited higher freezing levels compared to those raised in larger litters (7–12 pups), whereas unfamiliar mice showed the reverse relationship (Fig. 1d). Importantly, average litter size did not differ between any of these groups (Supplementary Table 1).

We further analyzed locomotor activity measurements obtained immediately after demonstrator foot-shock (measurement (M)1) or across the remainder of each trial (M2-12) (see Fig. 1a, top). Initial locomotor activity (M1) was equivalent across groups, indicating similar orientation responses for all observers; however, unfamiliar male mice failed to maintain freezing behavior throughout each conditioning trial (M2-12) (Fig. 1e). These findings suggest that deficits in the transmission of fear from unfamiliar mice were not a result of deficits in the capacity to detect sensory signals emanating from these mice. We further found that locomotor activity was lower in familiar compared to unfamiliar observers prior to conditioning, indicating freezing in response to the mere presence of a confined demonstrator conspecific (Fig. 1b, e). We expected to see an increase in this behavior during the acclimation period on subsequent conditioning days (i.e., conditioned freezing). To the contrary, freezing was absent in both familiar and unfamiliar male mice during acclimation on the second conditioning day (Cond2; see Fig. 1a, bottom), and familiar males in fact showed an increase in locomotor activity during acclimation across the three conditioning days (Cond1–3; Supplementary Fig. 6).

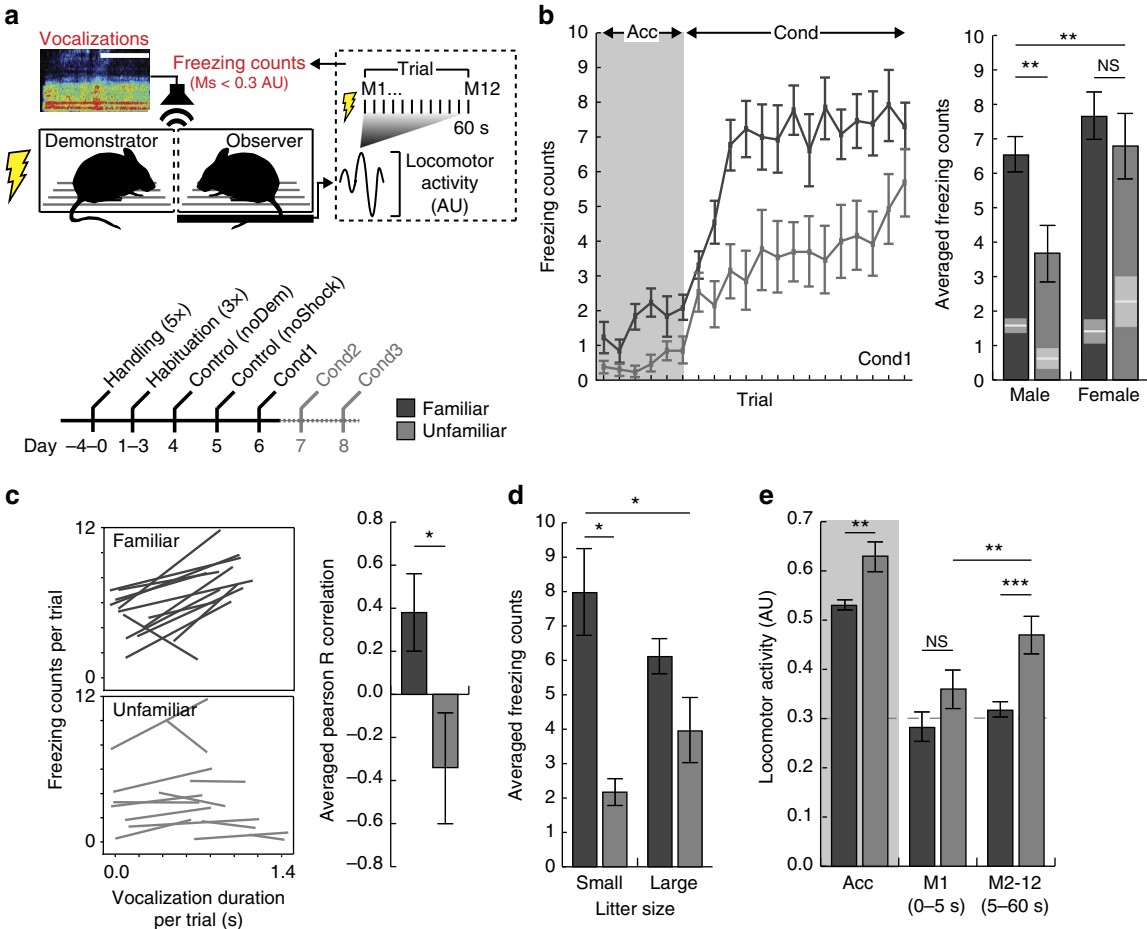

**Fig. 1** Unfamiliar male mice exhibit a reduction in socially transmitted fear. **a** Schematic of the socially transmitted fear paradigm (AU, arbitrary unit; M, measurement; scale bar, 100 ms), and schedule (Acc, acclimation; Cond, demonstrator conditioning). **b** Unfamiliar male observers (gray, $n = 13$) exhibited significantly less freezing compared to familiar male observers (black, $n = 13$) across demonstrator conditioning trials (repeated measure ANOVA: $F_{1,390} = 6.00$, $p < 0.0001$), and when averaged (two-tailed Student's t-test: $t(25) = 2.96$, $p = 0.008$). Familiar (black, $n = 10$) and unfamiliar (gray, $n = 10$) female mice did not exhibit significant differences in levels of freezing (two-tailed Student's t-test: $t(19) = 0.728$, $p = 0.48$) (ANOVA: $F_{3,45} = 5.55$, $p = 0.003$; effect of familiarity $F_{1,45} = 6.06$, $p = 0.018$; effect of sex $F_{1,45} = 7.83$, $p = 0.008$; familiarity x-sex interaction $F_{1,45} = 1.76$, $p = 0.192$). The gray box indicates the acclimation period prior to demonstrator conditioning. White lines and gray boxes within bars indicate mean and s.e.m. of freezing counts, respectively, during acclimation. **c** The relationship between observer freezing counts and demonstrator vocalization durations across conditioning trials in familiar and unfamiliar mice. Average correlations were significantly different (Fisher transformation, two-way Student's t-test: $t(24) = 2.25$, $p = 0.035$). **d** Socially transmitted fear differed between familiar (black; small: $n = 3$, large: $n = 10$) and unfamiliar (gray; small: $n = 2$, large: $n = 11$) male mice when the observer was reared within a relatively small litter (small: 2–6 pups; large: 7–12 pups) (ANOVA: $F_{3,25} = 3.69$, $p = 0.027$; effect of familiarity, $F_{1,25} = 10.28$, $p = 0.004$; effect of size $F_{1,25} = 0.001$, $p = 0.970$; familiarity x-size interaction, $F_{3,25} = 2.16$, $p = 0.156$; two-tailed Student's t-test: small litters only, $t(4) = 4.38$, $p = 0.035$). **e** Locomotor activity measurements recorded during acclimation, and over the first 5 s (M1) or thereafter (M2-12) demonstrator foot-shock trials were significantly different in unfamiliar observers (two-tailed Student's t-test: effect of measurement $t(25) = 3.70$, $p = 0.002$), and in unfamiliar compared to familiar observers at M2-12 (two-tailed Student's t-test: effect of familiarity $t(25) = 3.96$, $p = 0.001$), indicating less sustained freezing within trials. The dashed line indicates locomotor activity threshold used for freezing counts, as in **b**. Error bars represent s.e.m. *$p < 0.05$; **$p < 0.01$; ***$p < 0.001$; NS = not significant

**Repeated observation of a fearful demonstrator evokes escape.** Increased locomotor activity of familiar observer mice suggested a transition from freezing to escape behavior. Escape is an ethologically relevant social fear behavior that can be elicited in response to the communicated distress of a conspecific[34]. To test the inference that familiar observer mice exhibit escape behavior following extended exposure to demonstrator distress, we designed a novel socially induced avoidance paradigm (Fig. 2a, Supplementary Movie 2). In this paradigm, observers experienced demonstrator conditioning on two subsequent days: on the first (Cond (closed)), observers were confined to the demonstrator-containing side; on the second (Cond (open)), observers were allowed to escape to the opposite side. During acclimation on this second conditioning day, both familiar and unfamiliar observer mice exhibited significantly

more time on the non-demonstrator side compared to non-conditioned controls. However, only familiar observers maintained this behavior during conditioning trials (Fig. 2b). Although side preferences were equivalent for all groups at pre-test (measured prior to conditioning), only familiar male observers exhibited a significant preference for the non-demonstrator side at the post-test, measured on the day following the last conditioning day (Fig. 2c). Finally, a normalized score of avoidance indicated that only familiar observer mice exhibited a conditioned avoidance of the demonstrator-containing side (Fig. 2d).

**Intranasal oxytocin enhances social fear.** One putative molecular substrate of empathy is oxytocin, a neuropeptide conserved

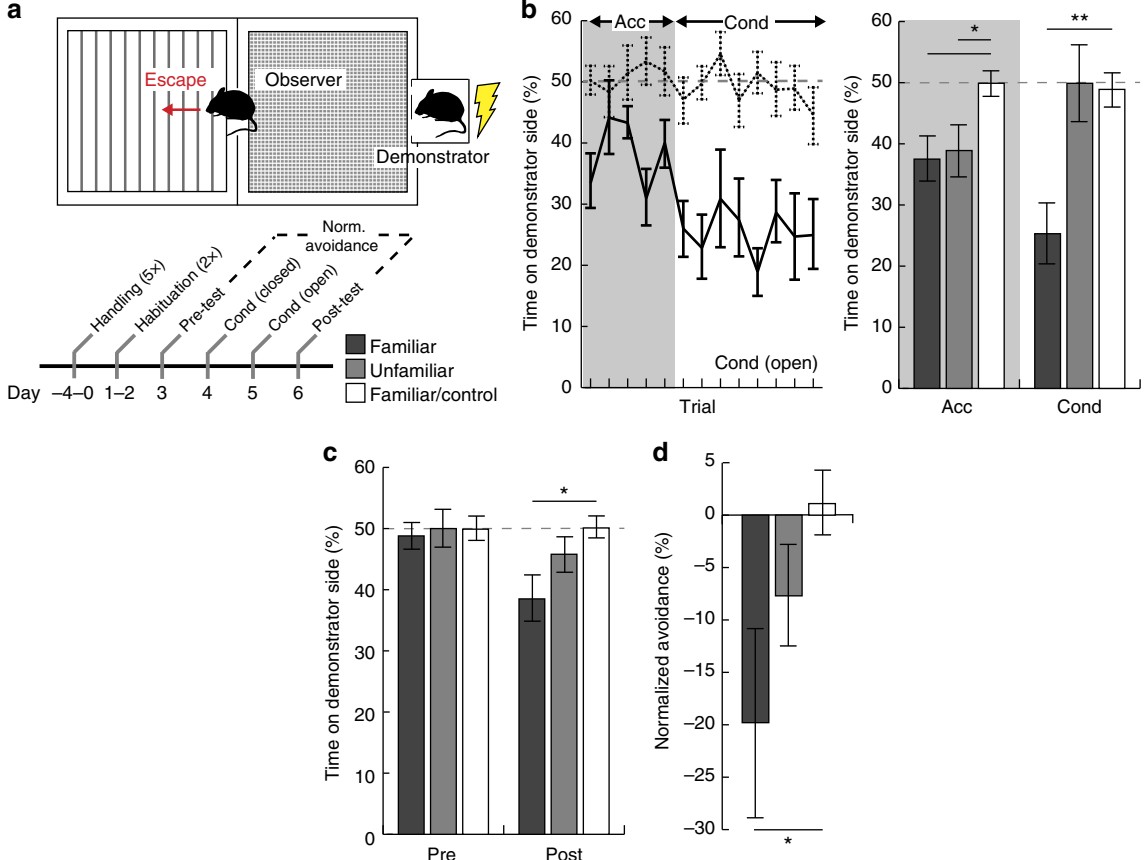

**Fig. 2** Familiar male mice exhibit escape behavior in a novel socially induced avoidance paradigm. **a** Schematic of the socially induced avoidance paradigm, and schedule (Acc, acclimation; Cond, demonstrator conditioning). **b** On the second conditioning day (Cond (open)), familiar mice (black line, n = 10) but not non-conditioned controls (dashed line, n = 12) escaped the demonstrator-paired context during acclimation and conditioning. The gray box indicates acclimation period prior to demonstrator conditioning. Both familiar (black, n = 10) and unfamiliar (gray, n = 8) observer mice exhibited initial escape behavior during acclimation (familiar vs. control: two-tailed Student's t-test: t(21) = 2.90, p = 0.011; Dunnett's post hoc: LSD = 0.021, p = 0.017; unfamiliar vs. control: t(19) = 2.32, p = 0.042; Dunnett's post hoc: LSD = −0.0004, p = 0.049), whereas only familiar mice maintained escape during demonstrator conditioning (familiar vs. control: two-tailed Student's t-test: t(21) = 4.09, p = 0.001; Dunnett's post hoc: LSD = 0.093, p = 0.001; unfamiliar vs. control: t (19) = 0.14, p = 0.889; Dunnett's post hoc: LSD = −0.14, p = 0.981). **c** Compared to non-conditioned controls (white, n = 12), familiar observers (black, n = 10) exhibited significant escape behavior at post-test (two-tailed Student's t-test: t(21) = 2.73, p = 0.017; Dunnett's post hoc: LSD = 0.025, p = 0.012), whereas unfamiliar mice (gray, n = 8) did not (two-tailed Student's t-test: t(19) = 1.27, p = 0.229; Dunnett's post hoc: LSD = −0.05, p = 0.488). **d** Familiar observer mice (black, n = 10) exhibited a significant difference in normalized avoidance scores compared to non-conditioned controls (white, n = 12) (two-tailed Student's t-test: t(21) = 2.17, p = 0.052; Dunnett's post hoc: LSD = 0.023, p = 0.029). Error bars represent s.e.m. *p < 0.05; **p < 0.01

through mammalian evolution. In humans, intranasal oxytocin bolsters emotional recognition[11] and empathy[12]. We therefore sought to strengthen socially transmitted fear in unfamiliar male mice using intranasal oxytocin administration (Fig. 3a). To assay oxytocin levels within the brain, we first collected cerebrospinal fluid (CSF) samples 30 min following intranasal administration (a sufficient period of time for oxytocin to reach the brain[35]) and quantified using an established ELISA procedure. Intranasal oxytocin successfully elevated CSF levels in behaviorally naive mice, thereby confirming the entry of oxytocin into brain CSF (Fig. 3b). We were unsuccessful in extracting and quantifying oxytocin from brain tissue (in contrast to CSF) using existing ELISA assays (Methods section). Hence, to validate the penetration of intranasal oxytocin into the mouse brain, we radiolabeled oxytocin and quantified its distribution across a number of brain regions 30 min following intranasal delivery. I[125]-labeled oxytocin was detected robustly within the blood and olfactory bulbs, and was distributed throughout the brain, including anterior olfactory nucleus, anterior cingulate cortex (ACC), nucleus accumbens, amygdala, and hypothalamus (Table 1). Using an acute regimen,

we then administered a single intranasal dose 30 min prior to demonstrator conditioning. For unfamiliar male mice, this regimen produced a significant enhancement of freezing compared to saline-treated controls (Fig. 3c). No discernible enhancement was detected following acute intranasal oxytocin administered to familiar male mice (Supplementary Fig. 7). Oxytocin signaling within the ACC has been recently implicated in rodent empathy behaviors[5,29]. We therefore investigated the effects of acute intranasal oxytocin on c-Fos, a marker of cellular activity, within the ACC. Using an immunohistochemical approach, we measured elevated c-Fos protein expression in the ACC, but not the neighboring primary motor region (M1), an area devoid of oxytocinergic inputs[36] (Fig. 3d).

Oxytocin has been shown to influence measures of explicit fear[15–18] and anxiety[19–21] in rodent models. Therefore, we also assayed the effect of single-dose intranasal oxytocin on classic measures of fear, namely, acquisition (freezing), expression (cue- and context-specific fear-potentiated startle (FPS)), and extinction. We did not note any differences in these measures when intranasal oxytocin was given acutely prior to conditioning

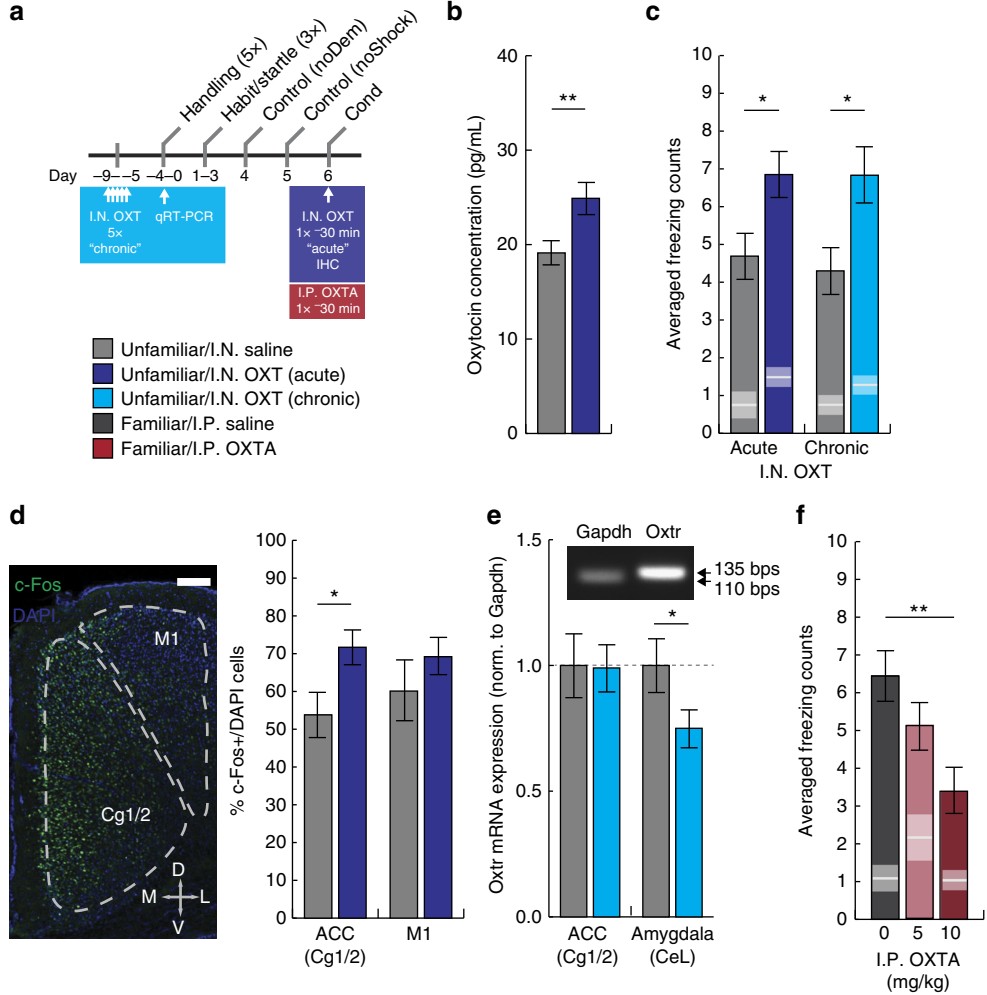

**Fig. 3** Oxytocin modulates socially transmitted fear behavior and neural mechanisms. **a** Schematic of oxytocin drug regimens and times of sample collection for qRT-PCR and immunohistochemistry (IHC; habit, habituation; cond, demonstrator conditioning; OXT, oxytocin; OXTA, oxytocin receptor antagonist). **b** In behaviorally naïve mice administered acute intranasal oxytocin (20 μg/kg; dark blue, $n = 10$), concentrations of the neuropeptide in CSF were significantly elevated compared to saline-treated controls (gray, $n = 10$) (two-tailed Student's t-test: $t(19) = 2.73$, $p = 0.015$). **c** Unfamiliar male observers given acute (20 μg/kg, single dose, 30 min prior; dark blue, $n = 12$; two-tailed Student's t-test: $t(23) = 2.49$, $p = 0.020$) or chronic (20 μg/kg, 5 daily doses; light blue, $n = 11$; two-tailed Student's t-test: $t(21) = 2.64$, $p = 0.016$) intranasal oxytocin exhibited a significant enhancement of freezing compared to saline-treated controls (gray, $n = 12/11$, respectively). White lines and gray boxes within bars indicate mean and s.e.m. of freezing counts, respectively, during acclimation. **d** Immunofluorescent labeling of c-Fos+ and DAPI cells within the anterior cingulate cortex (Cg1/2) and primary motor area (M1) (scale bar, 100 μm). Acute intranasal oxytocin significantly enhanced the percentage of c-Fos+/DAPI cells within the ACC (two-tailed Student's t-test: $t(9) = 2.36$, $p = 0.048$) but not M1 (two-tailed Student's t-test: $t(9) = 0.96$, $p = 0.369$) ($n = 5$ mice per group). **e** Chronic intranasal oxytocin significantly reduced oxytocin receptor (Oxtr) transcript expression in the lateral division of the central nucleus of the amygdala (CeL) (two-tailed Student's t-test: $t(19) = 2.08$, $p = 0.05$), but not the ACC (two-tailed Student's t-test, $t(19) = 0.06$, $p = 0.95$), assayed one day after completion of chronic oxytocin treatment. qRT-PCR amplicons of the Gapdh (110 bps) and Oxtr (135 bps) transcripts at expected sizes. **f** Familiar male observers systemically administered the oxytocin receptor antagonist L-368,899 hydrochloride (OXTA; I.P. single dose, 30 min prior; red, $n = 12$ per group) showed dose-dependent reductions in socially transmitted fear at 5 mg/kg and 10 mg/kg, compared to saline-treated controls (ANOVA: $F_{2,35} = 5.73$, $p = 0.007$). Error bars represent s.e.m. *$p < 0.05$; **$p < 0.01$

(Supplementary Fig. 8). Therefore, the effects of oxytocin on socially acquired fear were not secondary to alterations in the capacity for fear conditioning in general.

Duration of oxytocin exposure has been proposed to influence behavioral outcomes in both humans[37] and animal models[38,39]. Given that repeated administration of low doses of oxytocin can produce lasting effects on social interaction behavior[22] we employed a chronic regimen in which intranasal oxytocin was administered on five consecutive days, followed by a 10-day period without drug exposure (Fig. 3a). Similar to acute oxytocin, this regimen produced a significant enhancement of freezing in oxytocin-treated unfamiliar observers, compared to saline-treated

controls (Fig. 3c). These oxytocin effects could not be attributed to differences in the duration of vocalizations of the demonstrator mice (Supplementary Fig. 9) or to differences in acquisition of Pavlovian fear in general following chronic intranasal oxytocin administration to the observer mice (Supplementary Fig. 10).

Oxytocin stimulates a distinct population of neurons within the lateral division of the central nucleus (CeL) of the amygdala that express the G-protein-coupled oxytocin receptor (OXTR)[40]. In light of the established role of the amygdala in fear behaviors, alterations in OXTR expression in the CeL may be involved in the behavioral effects of chronic intranasal oxytocin. We therefore collected brain tissue from mice treated with chronic oxytocin

**Table 1 Concentrations of $I^{125}$-labeled oxytocin following intranasal administration**

| $I^{125}$-oxytocin  Concentration (μM) | Mean (SE)  (n = 8) |
|---|---|
| *Tissue/region* | |
| Blood | 1.94 (0.097) |
| Olfactory bulb | 1.46 (0.307) |
| Anterior olfactory nucleus | 0.40 (0.123) |
| Anterior cngulate cortex | 0.14 (0.038) |
| Frontal cortex | 0.08 (0.004) |
| Caudate/putamen | 0.06 (0.004) |
| Nucleus accumbens | 0.15 (0.052) |
| Septal nucleus | 0.07 (0.007) |
| Amygdala | 0.14 (0.027) |
| Hippocampus | 0.08 (0.011) |
| Parietal cortex | 0.08 (0.029) |
| Thalamus | 0.07 (0.003) |
| Hypothalamus | 0.22 (0.053) |
| Midbrain | 0.11 (0.016) |
| Pons | 0.12 (0.020) |
| Medulla | 0.12 (0.020) |
| Cerebellum | 0.08 (0.006) |

and measured levels of the Oxtr mRNA transcript in the amygdala and ACC. Compared to saline-treated controls, amygdala tissue recovered from oxytocin-treated mice showed a significant downregulation of the Oxtr transcript, whereas there was no difference in the ACC (Fig. 3e, Supplementary Fig. 11).

**Systemic oxytocin receptor antagonism reduces social fear**. To investigate the contribution of endogenous oxytocin signaling in socially transmitted fear, we systemically administered the non-peptidergic oxytocin receptor antagonist (OXTA) L-368,899 hydrochloride to familiar male mice. Administration of the OXTA 30 min prior to demonstrator conditioning (Fig. 3a) produced a dose-dependent reduction in freezing compared to saline-treated controls (Fig. 3d), but no differences in direct acquisition of Pavlovian fear in these same observer mice (Supplementary Fig. 12a). We further investigated whether levels of freezing measured as a result of socially transmitted fear correlated with directly acquired fear resulting from contextual Pavlovian conditioning. We predicted that these measures of fear, whether acquired socially or directly within the same mice, would involve common neural systems but would only be associated for mice with intact oxytocin signaling. Indeed, saline-treated familiar male mice exhibited a significant correlation between socially and directly acquired freezing levels, a relationship that was not recapitulated in OXTA-treated mice (Supplementary Fig. 12b). Collectively, these findings support the idea that socially and directly acquired fear are subject to individual differences along a common trait of fearfulness[41] and that oxytocin neurotransmission contributes exclusively to the former.

**DREADD activation of oxytocin neurons enhances social fear**. Circulating oxytocin levels appear to contribute to elevated socially transmitted fear in familiar male mice. To investigate the remediating effects of enhanced endogenous oxytocin signaling on socially transmitted fear behavior in unfamiliar mice, we utilized designer receptors exclusively activated by designer drugs (DREADDs) to chemogenetically activate oxytocinergic neurons. We injected a conditional rM3D(Gs)-encoding adeno-associated virus (Fig. 4a) bilaterally into the hypothalamic paraventricular nuclei (PVN) of Oxt-IRES-Cre/+ mice[42], for which Cre expression is restricted to oxytocin-producing neurons (Fig. 4b). In mice administered the DREADD ligand clozapine-N-oxide (CNO; I.P.) 90 min prior to killing, immunofluorescent imaging revealed co-

localization of rM3D(Gs) receptors and c-Fos+ neurons within the PVN (Fig. 4c), and the percentage of c-Fos+/DAPI cells was significantly elevated compared to mice administered saline (Fig. 4d). In unfamiliar male mice expressing rM3D(Gs) within oxytocinergic neurons, activation of DREADDs via CNO administered 30 min prior to demonstrator conditioning produced a significant enhancement of freezing compared to saline-treated mice or CNO-treated mice infected with a control, GFP-encoding virus (Fig. 4e). This DREADD-mediated enhancement in unfamiliar mice reached a level of socially transmitted fear commensurate with that seen in familiar mice (Fig. 1). There were no significant differences in demonstrator vocalizations (Supplementary Fig. 13), nor did DREADD activation cause differences in acquisition or retrieval of directly acquired fear (Supplementary Fig. 14).

## Discussion
The experiments described here demonstrate socially transmitted (i.e., observational) fear in mice. Similar to reports of emotional state-matching in humans[12,43], socially transmitted fear was seen to a greater extent in females than in males and was bolstered by administration of the neuropeptide oxytocin. We further showed using DREADDs the involvement of oxytocinergic neurons of the hypothalamus in expression of this phenomenon. Together with recent findings in prairie voles[29], these experiments affirm the essential role of oxytocin in rodent mental states akin to human forms of empathy. The mice are considerably less socially interactive than prairie voles as adults[44], but are especially amenable to genetic and genomic investigation. Thus, our findings are helpful in establishing the neural substrates of empathy, a capacity that is the glue of our social world[2], and deficits in which characterize several psychiatric and developmental conditions[2–4].

The form of empathy we studied was the tendency of one mouse to assume the emotional state of another. This phenomenon, referred to as emotional state-matching or affect sharing[1,45], was measured here as the social transmission of fear, homologous to fear contagion in humans[46]. This form of empathy was manifested initially as expression of freezing by the observer mouse. Freezing both coincided with, and extended beyond, foot-shock delivery to the demonstrator mouse, did not require previous experience with foot-shock, and correlated with the demonstrator's emission of audible distress vocalizations. When this procedure was repeated over several days the freezing response was replaced by increased activity and efforts at escape. Freezing and escape are incompatible behaviors (involving a cessation versus initiation of activity, respectively) and are prototypical expressions of fear across mammalian species[47–49]. Within naturalistic contexts, rats show a similar transition from passive freezing to active escape, depending on the proximity of the threat[47]. Moreover, observational or vicarious fear conditioning in rats is dependent on the vocalizations of the demonstrator[8]. Taken together, the exhibition and timing of these responses in mice strongly suggest that fear is transmitted to the observer via the vocalizations of the demonstrator.

Supporting the contention that social fear in mice and empathy in humans are homologous phenomena, a similar directionality of gender differences is seen in both species. Females are more empathic than males across several domains of human social interaction[43], and here we found stronger emotional state-matching in females than in male mice. A scarcity of previous reports of female superiority in empathy-like behavior in rats[50] and mice[51] likely reflects the fact that such studies have for the most part been conducted exclusively in males[5,7–9,52]. This gender difference may be a result of higher receptivity of female mice to the vocalizations of other mice, or of greater vocal

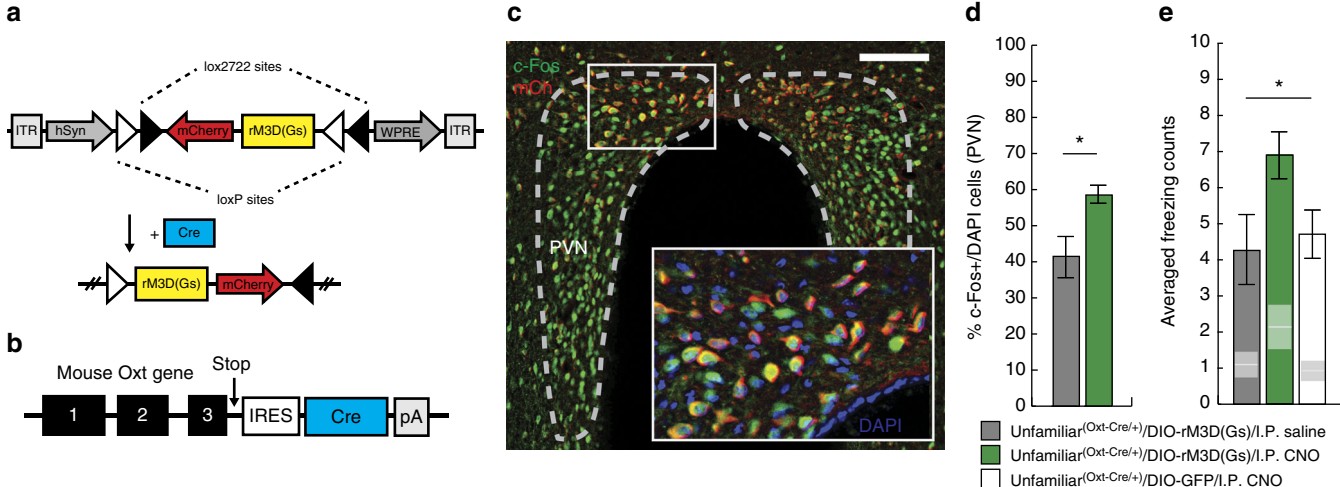

**Fig. 4** Chemogenetic activation of oxytocinergic neurons enhances socially transmitted fear in unfamiliar male mice. **a** Schematic of construct used for viral incorporation of the rM3D(Gs) receptor before and after Cre-dependent recombination. **b** Schematic of Cre recombinase transgene insertion at the mouse oxytocin gene. **c** Representative c-fos and mCherry immunoreactivity within the paraventricular nuclei (PVN; dashed outline) expressing rM3D(Gs) receptors (scale bar, 100 μm) and magnified view of co-labeled c-Fos+/mCherry-expressing neurons. **d** Quantification indicating significantly elevated percentage of c-Fos+/DAPI cells within the PVN of CNO-treated mice (3 mg/kg, I.P.; green, $n = 4$ mice) compared to saline-treated controls (gray, $n = 4$ mice) expressing the rM3D(Gs) DREADD (two-tailed Student's $t$-test: $t(7) = 2.65$, $p = 0.050$). **e** Administration of CNO (3 mg/kg, I.P.) 30 min prior to demonstrator conditioning enhanced freezing in unfamiliar male mice expressing rM3D(Gs) (green, $n = 10$) compared to saline-treated controls with rM3D(Gs) (gray, $n = 9$) and CNO-treated controls without rM3D(Gs) (white, $n = 10$) when averaged over conditioning trials (ANOVA: $F_{2,28} = 3.41$, $p = 0.048$). White lines and gray boxes within bars indicate mean and s.e.m. of freezing counts, respectively, during acclimation. Black error bars represent s.e.m. *$p < 0.05$

expressiveness among female demonstrator mice. Consistent with the latter possibility, female mice produce more high-frequency calls than do male mice[53], and we found a similar sex difference in the frequency of audible vocalizations during foot-shock.

We traced the origins of the lower level of observational fear in male mice to two factors. First, responsiveness of male mice was positively related to their familiarity with the conspecific in distress. This familiarity bias has been widely demonstrated in mice[5,31,52] and rats[50] within experimental settings and in ground squirrels within their natural environment[34], suggesting an evolutionarily conserved function. Evidence for a positive association between observer freezing and duration of vocalizations emitted from demonstrators implies this familiarity bias may be attributable, at least in part, to heightened recognition of auditory cues. Given that the current experiments did not disambiguate relatedness (i.e., siblings) from familiarity (i.e., cage-mates), this effect may also reflect a genetic predisposition for responding preferentially to relatives. Second, social transmission of fear between unfamiliar males depended on the size of the litter in which the observer mouse had been raised. That is, male mice raised in large litters were more responsive to an unfamiliar male than were those raised in small litters. This effect could result from several epigenetically mediated factors, including perinatal nutritional or hormonal environment, the quality of maternal care, and/or degree of sibling interaction[54]. The latter factor may be important in providing the experience of interacting with—and especially communicating with—a wider circle of conspecifics. Interestingly, 24 h of social isolation during adolescence has proved sufficient to reduce social transmission of fear in male mice[51]. Alternatively, it may be more stressful for mice to be raised with few siblings[55,56], and heightened stress has been proposed to underlie the reduced emotional state-matching among unfamiliar conspecifics[52]. Regardless of its underlying cause, our litter-size effect comports with the human literature showing that the number of siblings can influence measures of social cognition[57].

Having established reliable observational fear in mice as a form of empathy-like behavior we next set about investigating its neural substrates. Recent evidence has demonstrated a critical role for oxytocin in several forms of empathic behavior: maternal retrieval of mouse pups[58], as well as grooming of a stressed conspecific (consolation), and social transmission of fear in prairie voles[29]. We demonstrated the necessity of oxytocin signaling for social transmission of fear in mice by administering an oxytocin receptor antagonist and thereby reducing freezing in familiar observers. To demonstrate the role of the central oxytocin system in social transmission of fear, we then conditionally induced DREADDs expression in oxytocinergic neurons within the paraventricular nuclei (PVN) of the hypothalamus. This approach allowed us to stimulate oxytocinergic neurons in mice during an episode of observing the distress of an unfamiliar conspecific. Activation of DREADDs produced social transmission of fear in observer mice, showing that stimulation of PVN oxytocin neurons can lead to the elicitation of observational fear in circumstances where it would not otherwise be seen. Together, these findings indicate that the oxytocin system is both necessary and sufficient for the social transmission of fear.

The involvement of the oxytocin system in emotional state-matching is consistent with evidence in human subjects of the prosocial effects of oxytocin administered acutely, including increased emotional recognition[59] and empathy[12]. In human subjects oxytocin is typically delivered intranasally; however, the extent of its penetration to the brain has been uncertain[60]. Using a radiolabeling approach, we sought to validate the presence of intranasally administered oxytocin in various regions of the mouse brain. Although the I[125]-radiolabeling technique cannot distinguish between intact peptide and labeled fragments of the peptide, detection of intact oxytocin in CSF by ELISA argues against the latter possibility in the current study, consistent with previous chromatography analyzes (see ref. [61] for discussion). In particular, I[125]-labeled oxytocin was detected at high levels within the olfactory bulb and anterior olfactory nuclei, consistent with a mechanism of entry via olfactory nerves of the lamina propria[61]. More relevant to our study, we also observed elevated levels within the ACC, where we noted enhanced c-Fos

immunoreactivity, amygdala, where we noted altered expression of the oxytocin receptor, and hypothalamus, where we chemogenetically modulated oxytocinergic neurons. This distribution pattern agrees with previous studies using other intranasally delivered peptides, largely in rats[62,63], and cerebral brain flow effects following intranasal oxytocin in mice[64]. The breadth of its distribution in structures rich in oxytocin receptors and implicated in other empathy-related behavioral domains may suggest that exogenously applied oxytocin affects empathic behavior via its actions on a network of neural substrates, such as the hypothalamus, ACC, and amygdala[14,36,65].

Utilizing the intranasal administration route, we found that exogenous oxytocin fully rescued social transmission of fear in unfamiliar male mice. Of greater clinical relevance[66], we saw similar facilitation after an extended oxytocin dosing protocol that has been found to enhance social behavior in mice[22]. Interestingly, long-lasting benefits on social cognition have also been documented using intranasal oxytocin treatment for autism spectrum disorders[67]. Since our last dose of oxytocin had been administered a full 10 days prior to behavioral testing, the facilitation of social transmission of fear must have occurred through persistent neuroadaptive changes rather than through the direct effects of the drug. Indeed, we found that chronic oxytocin downregulated transcription of the oxytocin receptor in the lateral division of central nucleus (CeL) of the amygdala, a critical locus for the expression of fear conditioning[68]. This effect may be functionally similar to the internalization of receptors observed upon extended oxytocin overstimulation within the central nucleus of the amygdala[69]. In contrast, the prosocial effects of acute oxytocin appear to involve the ACC, consistent with evidence linking this region (particularly oxytocin signaling within it[29]) to empathy-related processes across mammalian species[5,70]. We found no long-term change in oxytocin receptor transcription in the ACC, suggesting a dissociation between the loci of acute and chronic oxytocin effects on social transmission of fear. The ACC and CeA are two primary targets of hypothalamic oxytocinergic neurons[36], and oxytocin enhances the functional connectivity between these regions[14,65]. Furthermore, observation of a distressed conspecific enhances overall synaptic transmission between the ACC and amygdala[71]. Hence, interplay within oxytocin-modulated hypothalamic-cingulate-amygdalar circuitry likely plays a critical role in the social transmission of fear[14].

It is notable that in several experiments, we found no effects of either increasing or decreasing oxytocin activity acutely on Pavlovian fear conditioning. This is an important control in order to show that oxytocin affects the social transmission of fear rather than the acquisition or expression of fear itself. We similarly found no effect of our chronic oxytocin regimen on fear conditioning. This suggests that downregulation of oxytocin receptors in the CeA is specific to neurons that are also activated by socio-emotional cues, perhaps via the ACC[5]. More generally, the absence of any effect of either acute or chronic oxytocin on fear conditioning may require us to re-evaluate the role that oxytocin plays in fear and anxiety. The results of a number of other studies have suggested that oxytocin is anxiolytic[19–21] or attenuates measures of fear conditioning[15–18]. In contrast, we showed that oxytocin or an oxytocin receptor antagonist enhances, or blocks, socially transmitted fear, respectively. A plausible resolution to these apparently paradoxical effects is that the reductions in fear and anxiety previously seen in rodent studies were secondary to the prosocial effects of oxytocin. In both humans and rodents, fear and anxiety can be buffered by the presence of another nonfearful individual[72,73]. Thus, the presence of a home-cage conspecific at the time that oxytocin is administered could have the incidental effect of ameliorating subsequent acquisition of fear. When this factor is controlled for, as in this study (and see ref.

[28]), modulatory effects of oxytocin on fear acquisition are not seen.

Given the centrality of human interpersonal relations to well-being—the rewarding nature of social interaction and the stressful consequences of its absence—characterizing the neural bases of empathy represents an important step for understanding the development and manifestation of both adaptive and maladaptive mental states. Fear contagion in humans correlates positively with several other measures of empathy, including those with no obvious murine equivalent[74]. Thus, observational fear would appear to be well suited as a behavior for investigating genetic, developmental, and neural substrates of empathy in mice. In this study, we have demonstrated the importance of the brain oxytocin system in the social transmission of fear among mice. With this knowledge, the stage is set for dissecting the circuitry and neurophysiology of empathy through development and into adulthood. This may lead to improved treatments for the prevention and amelioration of cases of psychiatric and developmental diseases, such as autism spectrum disorders, in which reduced empathy processes are a prominent feature.

## Methods

**Animals and genotyping.** C57/B6 mice were bred and maintained according to the US National Institutes of Health guidelines for animal care and use and Institutional Animal Care and Use Committee of the University of Minnesota—Twin Cities. Male or female mice were pair-housed with same-sex littermates from the time of weaning and tested in behavioral experiments at 8–12 weeks of age. Mice were provided food and water ad libitum except during experimental testing, and housing lights were maintained on a 12:12 h light/dark cycle. Oxt-IRES-Cre/+ transgenic mice (Jackson Laboratories #024234) were genotyped using the following primers: 5′-TTTGCAGCTCAGAACACTGAC-3′ (F); 5′-ACACCGGCCT-TATTCCAAG-3′ (Mut-R), 5′-AGCCTGCTGGACTGTTTTTG-3′ (WT-R).

**Drug administration.** Oxytocin (Sigma-Aldrich #O3251) was dissolved in sterile saline and aliquots were frozen at −80 °C. The oxytocin antagonist (OXTA, L-368,899 hydrochloride, Tocris Bioscience #160312-62-9) was dissolved in sterile saline to make a 0.625 mg/mL working solution, and aliquots were frozen at −20 °C. For intranasal oxytocin, mice were briefly anesthetized with isoflurane and quickly administered oxytocin (20 µg/kg) or saline at 5 µL per nostril. Chronic oxytocin was administered at approximately the same time for five consecutive days using the same dose (20 µg/kg). For the systemic oxytocin receptor antagonist (OXTA), mice were briefly anesthetized with isoflurane and quickly administered the selective OXTA L-368,899 hydrochloride (5 or 10 mg/kg, I.P.). Systemically administered L-368,899 hydrochloride is known to reach the brain, where it preferentially binds with oxytocin receptors[75].

**CSF collection and oxytocin measurements.** Cerebrospinal fluid (CSF) was obtained according to techniques described previously[76]. Mice were deeply anaesthetized using Beuthenasia (200 mg/kg, I.P.) at 30 min following intranasal oxytocin administration. Mice were then placed into a stereotaxic frame and a longitudinal cut was made along the posterior neck. Subcutaneous tissue and muscles were exposed, sectioned along the rim of the occipital bone, and then removed laterally to reveal the clear arachnoid membrane overlying the cisterna magna. A glass capillary was pulled to ~5 mm diameter and inserted horizontally into the cisterna magna. CSF samples were obtained between 1 and 3 pm, and typically 2–3 µl of total CSF was collected. Samples contaminated with blood were discarded. Samples were immediately frozen at −80 °C. Oxytocin quantification was completed using standard ELISA techniques and according to the manufacturer's protocol (#ADI-153A, Enzo Life Sciences, Inc.). Briefly, CSF samples were added to an equivalent volume of 0.1% trifluoroacetic acid (TFA) in water, then applied to a C18 spin column (Thermo Scientific, Inc.) and washed with 0.1% TFA. The extracted oxytocin sample was eluted using a 95% acetonitrile/5% of 0.1% TFA solution and evaporated at 4 °C. Samples were then reconstituted in assay buffer and run in duplicate on a spectrophotometer (Multiskan EX, Thermo Scientific, Inc.) at 405 nm. Optical density values were converted to concentration using the manufacturer's standard curve. Neither the aforementioned ELISA kit (Enzo Life Sciences, Inc.) nor a second kit (#K048, Arbor Assays, Inc.) proved effective for extracting and quantifying oxytocin contained within brain tissue.

**Quantification of I$^{125}$-oxytocin brain distribution.** Oxytocin (Sigma-Aldrich #O3251) was custom I$^{125}$-labeled using a lactoperoxidase method (PerkinElmer, Inc.) and validated at an activity level of 81.4 TBq/mmol (1945 µCi/µg). Radiolabeled oxytocin was diluted with unlabeled oxytocin and delivered to adult male mice at 2 mg/kg (30 µCi) using a similar dosing approach as detailed above. 30 min

following administration, mice were killed, blood collected, and perfused through the descending aorta with 20 mL of 0.9% sodium chloride, followed by 120 mL of 4% paraformaldehyde in 0.1 M Sorenson's phosphate buffer (10 mL/min). The brain was removed from the skull and olfactory bulbs dissected away from the brain. Serial (2 mm) coronal sections of the brain were made using a mouse brain matrix (Braintree Scientific). Microdissection of specific brain regions was performed on coronal sections. Each tissue sample was collected into a preweighed 5 mL tube, and the wet tissue weight was determined using a microbalance (Sartorius MC210S). Radioactivity in each tissue sample was then determined by gamma counting in a Packard Cobra II Auto Gamma counter (Packard Instrument Company). Concentrations were calculated using the half-life degradation of I[125], specific activity determined from standards, dosage of administered solution, counts per minute in the tissue following subtraction of background radioactivity, and tissue weight in grams.

**Socially transmitted fear paradigm**. Paired home-cage mice were randomly designated as observer or demonstrator prior to the start of testing. Observer mice were acclimated to handling by an experimenter for five consecutive days (5 min per day), followed by habituation to the testing context and acoustic startle over three consecutive days (see Acoustic Startle and Pavlovian Fear Conditioning Methods section). All social fear sessions commenced with a 5 min acclimation period following placement of the same-sex observer-demonstrator pair into adjacent Plexiglas cages. Activity measurements were recorded from the observer animal every 5 s using a load cell transducer, amplifier and computer equipped with Advanced Startle software (Med Associates, Inc.). The US was a 0.8 mA, 1.5 s scrambled foot-shock delivered through the bars of the cage to the demonstrator. These parameters were piloted to elicit reliable auditory and visual signs of distress (i.e., vocalizations and thigmotaxis/freezing behavior, respectively) from demonstrators. The conditioning protocol was composed of 15 trials, each containing 12 activity measurements following the US (60 s inter-trial interval). Prior to demonstrator conditioning days, observer mice underwent two consecutive days of control conditioning experiments: the first without inclusion of the demonstrator and with the US (noDem); the second with the demonstrator and without the US (noShock). Non-specific freezing during these experiments was assessed over "conditioning" trials (excluding acclimation period). Observer mice did not have experience with foot-shock (i.e., "priming") prior to demonstrator conditioning (cf. [7–9]). Freezing behavior was calculated as the number of measurements per trial (12 total) below a 0.3 AU activity threshold (see Fig. 1 and ref. [77]). Videos of observer mice were also recorded during conditioning sessions using infrared web cameras. Videos were scored for observer freezing behavior by a treatment-naive experimenter using Button Box 5.0 (Behavioral Research Solutions, LLC) (Supplementary Figs. 2, 4). Demonstrator freezing during conditioning was measured in a randomly selected subset of videos of male familiar and unfamiliar pairs (Supplementary Fig. 4). In a separate cohort, cued and contextual fear-potentiated startle was analyzed in demonstrator mice using the socially transmitted fear conditioning protocol (Supplementary Fig. 1).

During conditioning sessions, vocalizations were collected using a high-frequency microphone, data acquisition hardware (UltraSoundGate 416 H; Avisoft Bioacoustics, Inc.), and recording software (Recorder USGH; Avisoft Bioacoustics). For analysis, recordings were processed using a custom-built R program. In brief, spectrograms were generated from raw WAV files and band pass filtered (10–110 kHz). A time-varying parameter, spectral purity, was computed as the fraction of total power within a single-frequency bin, and filtered over 8 ms. Identification of a vocalization bout was based on three parameters: minimal length (>5 ms), spectral purity (>0.15), and holding time (<10 ms), defined as the threshold time in which separate bouts are merged. Total duration was computed by summing the vocalizations identified across the entire conditioning session. Vocalizations were presumed to be elicited from demonstrators based on coincidence with foot-shock onset/offset and previously identified characteristics of distress vocalizations elicited by foot-shock[7]. An experienced experimenter confirmed the accuracy of automated call detection using the generated spectrogram. Vocalization recordings were not collected for one familiar male, one unfamiliar female, and two unfamiliar males with acute intranasal saline due to technical error.

**Socially induced avoidance paradigm**. The socially induced avoidance paradigm represents a novel method for studying escape-specific social fear behavior in mice. All experimental days entailed a 20 min testing session within a conditioned place preference arena (20 × 25 × 45 cm) containing both tactile (bars vs. grids) and visual (vertical lines vs. none) cues at 50 lux. The demonstrator mouse was confined to a small cage placed within a window on one side of the arena (counterbalanced). On two consecutive days, observer mice were habituated to the arena without inclusion of the demonstrator. The following day (pre-test), a demonstrator mouse was introduced into the adjacent conditioning cage and the observer was allowed to freely explore. On the first day of demonstrator conditioning (Cond (open)), the observer mouse was confined to the same side as the demonstrator. After 5 min acclimation, the demonstrator was exposed to foot shocks (1.5 s, 1 mA scrambled current per 1 min; 15 total) using the Advanced Startle software (Med Associates, Inc.). On the second day of demonstrator conditioning (Cond (closed)), the observer mouse was allowed to explore the entire arena while the demonstrator was conditioned using the same protocol. On the final day of testing (post-test),

observer mice were tested using a procedure identical to the pre-test. A normalized avoidance score was calculated as the change in percentage time spent on the demonstrator-containing chamber between pre-test and post-test: [post-time—pre-time)/(pre-time)]. Video footage was collected during all sessions using infrared web cameras and scored over 2-min bins by trained, group-naive experimenters using Button Box 5.0 (Behavioral Research Solutions, LLC).

**Acoustic startle and Pavlovian fear conditioning**. Acoustic startle and Pavlovian fear conditioning experiments were conducted in four ventilated, sound-attenuated acoustic chambers, each equipped with two speakers, a Plexiglas startle cage and a load cell transducer system (Med Associates, Inc.). The transducers are connected via an amplifier to a computer equipped with Advanced Startle software (Med Associates, Inc.). Startle sessions commenced with 5-min acclimation to the context in which no stimuli were presented, followed by 50 pseudo-randomized startle-eliciting noise pulses at 70, 80, 85, 90, and 100 dB (30 s ITI). Startle was recorded over a 500 ms time bin occurring 300 ms prior to pulse onset to 200 ms after pulse onset. Startle amplitude was measured as maximum to minimum peak amplitude of the sinusoidal response.

Pavlovian fear conditioning experiments were conducted using the same chambers as acoustic startle. On the first day (pre-test), individual mice were introduced into the testing cage, acclimated for 5 min, and presented with 20 "leader" startle stimuli (30 s ITI) in order to obtain baseline startle reactivity levels. Then, 24 additional startle stimuli were presented in a pseudorandom order, 12 with and 12 without the CS present. On two subsequent conditioning days, mice were re-introduced and acclimated for 5 min, followed by presentation of 15 CS-US pairings. The CS was a 30 s, 12 kHz, 70 dB explicit tone. The US was a 0.6 mA, 1 s (for observers; Supplementary Figs. 8, 10, 12, and 14) or a 0.8 mA, 1.5 s (for demonstrators; Supplementary Fig. 1) scrambled foot-shock delivered through the bars of the cage. On three subsequent days, mice underwent fear extinction during which they were presented with 20 "leader" startle stimuli (30 s ITI), followed by 24 additional startle stimuli (same ITI) without the CS. On the last day (post-test), the identical protocol was conducted as on the pre-test (i.e., with inclusion of the CS). Similar to the social fear experiment (see above), freezing during acquisition was calculated as the number of measurements per trial (12 total) below a 0.3 AU activity threshold and averaged across conditioning trials. Contextual fear-potentiated startle (FPS) was calculated as percent change in startle between averaged "leader" startle stimuli from pre-test to the first day of extinction (for observers) or post-test (for demonstrators). Cued FPS was calculated as percent change in startle between trials in which the CS was present and leader trials.

**Quantitative RT-PCR**. Total RNA was obtained from littermate-matched, experimentally naive male mice on the day following (18–24 h) the last chronic intranasal treatment (oxytocin or saline) (Fig. 3a). The tissue was dissected bilaterally from the whole amygdala and anterior cingulate cortex (ACC) using a mouse brain atlas reference, then quickly submerged in RNA Later (Qiagen) and frozen at −80 °C. Isolated RNA was processed for DNAase digestion and reverse-transcribed using the GoScript reverse transcriptase (Promega) according to the manufacture's manuals. The cDNA was diluted 1:7 before preparing the qRT-PCR reaction, which was then run using the GoTaq qPCR master mix (Promega). Oxytocin receptor (Oxtr) mRNA primers were chosen from previous work[78] and purchased through the University of Minnesota's Genomics Center. Primers were as follows: 5′-CCGCACAGTGAAGATGACCT-3′ (forward) and 5′-AGCATGGCAATGAT-GAAGGCAG-3′ (reverse). Mouse glyceraldehyde-3-phosphate dehydrogenase (Gapdh) mRNA was used as the endogenous control. For each sample, a duplex PCR reaction was set-up containing a target gene-primer set and a Gapdh-primer set. We conducted three replicate PCR reactions for each sample. The reactions were incubated in a 96-well plate at 95 °C for 10 min, followed by 45 cycles of 95 °C for 15 s and 62 °C for 1 min on the CFX96 Real-Time System (BioRad, Inc.). The relative levels of gene expression were determined according to the standard curve methods described in the BioRad company manual. The expression value of the target gene in each well was first normalized by the expression value of the Gapdh. The median of three repeated reactions was used to represent the relative quantity of the target gene.

**Chemogenetics**. Standard aseptic stereotaxic surgical procedures were utilized. The AAV2-hSyn-DIO-rM3D(Gs)-mCherry (~10[12] virus particles per mL; Addgene plasmid #50458) and AAV2-hSyn-DIO-GFP (~3[12] virus particles per mL; Addgene plasmid #50465) viruses were produced by the University of North Carolina Vector Core Facilities. Male Oxt-IRES-Cre/+ mice (7–8 weeks old) were sedated with a ketamine:xylazine cocktail (100:10 mg/kg) and bilateral burr holes were created above the paraventricular nuclei (PVN) of the hypothalamus. A Hamilton syringe was loaded with virus and slowly lowered to the PVN (AP: −0.85 mm; ML:±1.1 mm; DV: −5.4 mm; 10° angle). A total of 0.5 microliters was injected bilaterally at 0.1 μL/min, and the needle was left at the injection site for 10 min to allow for virus absorption. Animals were provided 500 μL saline (S.C.) post-operatively and 5 mg/kg Ketofen (S.C.) daily for three days, then allowed a minimum of 3 weeks recovery before commencement of behavioral experiments. Mice were then either injected with CNO (3 mg/kg, I.P.) or saline 30 min prior to social fear conditioning or

Pavlovian fear conditioning. For c-Fos immunohistochemistry, mice were administered CNO (3 mg/kg, I.P.) or saline 90 min prior to killing.

**Histology and c-Fos immunohistochemistry.** All animals used for chemogenetics experiments had their brains removed and processed to confirm viral injection location. Mice without Pavlovian fear conditioning experience were used for quantification of c-Fos+ neurons. Mice were deeply anaesthetized using Beuthenasia (200 mg/kg, I.P.) and transcardially perfused with PBS followed by ice-cold 4% paraformaldehyde in PBS. Brains were fixed overnight in 4% paraformaldehyde and 30% sucrose in PBS then sliced at 40 micron thickness using a vibratome (Leica VT1000S). Free-floating coronal sections containing the PVN hypothalamus were incubated for 3 h in blocking solution containing 0.2% Triton X-100, 2% normal horse serum, and 0.05% Tween20. Sections were then incubated for 72 h with primary antibodies (rabbit anti c-Fos 1:1000, Santa Cruz; mouse anti-mCherry 1:1000, Abcam), washed six times, and then incubated overnight with secondary antibodies (donkey anti-rabbit Alexa Fluor 488 IgG 1:1000, Life Technologies; goat anti-mouse Alexa Fluor 647 IgG 1:1000, Abcam). Slides were washed and counterstained for 20 min with 4′,6-diamidino-2-phenylindole (DAPI) (1:50,000, Life Technologies), and mounted. Fluorescent images were acquired on a Zeiss LSM 710 scanning confocal microscope using a ×20 objective, and analysis performed using ImageJ software. For chemogenetics experiments, quantification of c-Fos+ neurons was performed across eight slices containing the PVN (~−0.85 mm to bregma) by a treatment-naive experimenter. For intranasal oxytocin experiments, quantification of c-Fos+ neurons was performed across five slices containing the Cg1/2 and M1 regions (~+1 mm to bregma) by a treatment-naive experimenter.

**Statistics.** No statistical methods were used to predetermine sample sizes, although sample sizes were consistent with those from previous studies of this kind[7,9,29]. The difference between familiar and unfamiliar male mice freezing in the socially transmitted fear paradigm replicated several times in our laboratory (e.g., Figs. 1b, 3c, f and 4e). Data were normally distributed; therefore, parametric tests were used to compare groups. Two group comparisons were analyzed by unpaired two-tailed Student's t-test; three group comparisons were analyzed with one-way ANOVA. Dunnett's post hoc test was used for data in Fig. 2. Significance was set at $P < 0.05$ for all tests. Grubb's test was used to identify and remove outliers from socially transmitted fear experiments (using acclimation locomotor activity >2 standard deviations from group mean). Four mice were removed from analysis due to escaping the testing apparatus in the socially induced avoidance paradigm. One mouse was removed from analysis due to viral injection misplacement in chemogenetic experiments. Two mice were removed from analysis due to poor perfusion following I$^{125}$-oxytocin administration. Videos of demonstrator freezing were randomly selected from data in Fig. 1b for Supplementary Fig. 4. Slices used for c-Fos immunohistochemistry in Figs. 3d, 4d were randomly selected. Detailed statistics are provided within figure legends. All data are displayed as mean ± s.e.m.

**Data availability.** The data that support the current study are available from the corresponding author upon reasonable request.

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

## Acknowledgements

This research is supported by the Autism Initiative (University of Minnesota) (J.C.G.) and the University of Minnesota's MnDRIVE Predoctoral Neuromodulation Fellowship (M.T.P.). The authors would like to thank Aidan Peterson and Mike O'Connor for confocal imaging assistance; Amy Young, Parker Roy, and Jewoo Seo for assistance with data collection and video analysis; Michael Saxe and Yan-Ping Zhang for ultrasonic vocalization analysis code, Anders Asp and Erin Larson (MnDRIVE Optogenetics Core, University of Minnesota) for chemogenetic technical assistance, Aleta Svitek and Katherine Faltesek for radiolabeling technical and analytical assistance, and Patrick Rothwell, Mark Thomas, Suma Jacob, Bruce Kennedy, Brian Sweis, and Tatyana Matveeva for helpful discussions of the manuscript.

## Author contributions

M.T.P. designed, conducted, and analyzed experiments, and wrote the manuscript; I.I.G. guided experiments and reviewed and edited the manuscript; L.R.H. guided experiments; J.C.G. designed and guided experiments, and wrote the manuscript. M.T.P. and J.C.G. have read and approved the final manuscript. Our close colleague I.I.G. passed away prior to submission of the final manuscript.

## Additional information

**Competing interests:** The authors declare no competing financial interests.

