## [Peer Review File · Nature Communications]

Reviewers' comments:

Reviewer #1 (Remarks to the Author):

In this study, the authors examine the effect of oxytocin (OXT) on socially-transmitted behaviours in mice. Socially transmitted behaviours in rodents have been shown several times, both in rats and mice, and species differences as well as gender effects have been previously reported. However, all these studies remained at the behavioural level. The main result, demonstrated using both OXT pharmacology and pharmacogenetic manipulations, shows that higher oxytocin is associated with short and long-term enhancement of socially transmitted freezing. There have been a variety of experiments showing modulation of social behaviour in mice by OXT, but this result is to my knowledge novel because it was never shown that OXT can modulate socially transmitted behaviours.

The control experiments demonstrate that it is not due to direct effects on cue- or context-dependent fear conditioning, indicating that it is a social effect. The authors try to tie these effects to human empathy. The main weakness of the study is that the authors do not reveal what is being transmitted from Demonstrator (Dem) to Observer (Obs), hindering to some degree the interpretation of the results. The authors claim the mice are 'state matching' (which they equate with empathizing) but there is very little data documenting the behaviour of the Dem at all (only vocalizations). All in all this could be a very interesting study, but I have reservations with the interpretation of the experiments that I wish the authors to address before I would recommend publication.

Major points

1. The authors do not document what the Dem are doing during the paradigm, leaving unclear what is being transmitted between animals and complicating the interpretation of these effects as 'empathy'. The authors refer to what the Dem is doing sometimes as 'displaying distress', which seems fair (and loose), but other times they say that it is showing Pavlovian fear conditioning. Yet there is no documentation of this. In fact, and maybe I am not fully understanding the paradigm, but if indeed the Dem is being fear conditioned to the context, shouldn't the animal exhibit fear responses already during the acclimation period of Cond2? If yes, shouldn't those responses also be transmitted to the Obs? In Fig.1f, we see that the Obs is not freezing during acclimation period of Cond2 and Cond3. This either means that the Obs is not really responding to freezing per se (but we do not have any correlation of behaviour between the Dem/Observer pair besides vocalizations) or that the Dem did not learn. The only documentation shows that the observer is sensitive to the duration of vocalization (Fig. 1c) and that this measure does not change with OTX experiment – this is solid. But these experiments do not establish how much of the social effect on observer freezing is accounted for by this measure and how other modes of transmission may contribute. Therefore, I would like to have the report of the Dem's behaviour.
2. The OTX experiments have adequate but not very thorough controls. I would have liked to see the authors perform the manipulation experiments on both familiar and unfamiliar observers. Additionally, since CNO can have physiological effects, a control for CNO injection on Sham surgery animals would be helpful.
3. The effect of litter size, which is mainly a side-point, appears to be a post-hoc analysis. I wonder if this might be a statistical artefact resulting from checking multiple possible data splits which were not reported. If the authors wish to support this result a retest would greatly enhance confidence in this result.
4. The paradigm includes not only shock conditioning, which is the main focus, but also startle. This is only revealed in the Supplementary info and is not really described in the main text and

methods. I found this confusing. I wonder what the effects of the various conditions were on this second paradigm. The authors do show that OTX increases startle reactivity (Fig S6a) but don't comment on it. This result seems to be in disagreement to the main result, which is increase in freezing responses. I also wonder if there are any interactions between paradigms. Overall, I see why the authors would want to simplify the story, but I think they cannot keep this can of worms partly open.

5. SCPA experiment seems to distract from the main story. This paradigm which involves 'escape' is not as well documented. As it is an active response, it may have more to do with the reactivity assay than the fear assay, but this was not clearly described.

Minor points

1. The authors should mention two studies in rodents, Atsak et al, Plos1 2011 and Pereira et al, Current Biology 2012. The first study shows very similar results to the present study and the second one is a clear example of species difference. In the case of rats, prior experience is necessary for the exhibition of social transmission of fear.

2. Fig 1e – The authors should show the baseline movement here in order to allow comparison. Otherwise it is hard to know whether there is an absolute effect across both groups at M1. This seems to be true from Fig 1f.

3. I would like to see a more thorough discussion of the main result with OXT administration. According to Gozzi et al, Neuron, OXT administration leads to decrease in freezing response due to a direct effect on CeL cells that inhibit CeM (blocking freezing). However, in this paradigm OXT leads to an increase in freezing response on the Obs. How do the authors reconcile these two effects? Gozzi et al show the effect on a non-social paradigm. How do you think these two treatments can lead to disparate outcomes?

4. In general, I would appreciate a better presentation of the results, it is hard to compare locomotion activity with freezing counts, when those terms sometimes are almost used interchangeably.

Reviewer #2 (Remarks to the Author):

The manuscript by Pisansky and colleagues investigates the role of oxytocin in a model of mouse empathy. The study quite interesting and well-performed.

There are only a few items for the authors to address:

1. One can question whether empathy is the best description for mice. Emotional contagion provides an alternative explanation for this matched state and does not require Theory of Mind, which has not been documented in mice. This is a fundamental criticism of the terms used in this manuscript and could shift the scope of this research. It should be noted though that empathy and emotional contagion are not mutually exclusive and yet distinctions should be made, when appropriate. This could be discussed.

2. The authors note that oxytocin receptor expression in the ACC are not affected by chronic nasal administration of oxytocin and conclude a distinction in the role of oxytocin in acute vs chronic regulation of ACC activity as it related to empathy. However, it should be noted that the authors made no attempt to confirm that nasal administration of oxytocin affects local concentrations of oxytocin or oxytocin-related function in the ACC. Thus, the absence of change in the ACC could be

a product of the manipulation used.

3. The DREADD manipulation is not very conservative, given the wide-spread of oxytocin projections in the brain that originate from the PVN, including those involved directly in fear conditioning. Could the authors discuss or specify the scale of effect that this manipulation had on brain function?

4. The authors mention a couple of times that oxytocin has no role in fear learning. However, see, at least:

- Pagani, J. H., Lee, H.-J., & Young, W. S. (2011). Postweaning, forebrain-specific perturbation of the oxytocin system impairs fear conditioning. *Genes, Brain, and Behavior*, 10(7), 710–719.

and

- Modi, M. E., Majchrzak, M. J., Fonseca, K. R., Doran, A., Osgood, S., Vanase-Frawley, M., et al. (2016). Peripheral Administration of a Long-Acting Peptide Oxytocin Receptor Agonist Inhibits Fear-Induced Freezing. *The Journal of Pharmacology and Experimental Therapeutics*, 358(2), 164–172.

5. An important study of “empathy” in rodents not cited is:

Smith, A. S., & Wang, Z. (2014). Hypothalamic oxytocin mediates social buffering of the stress response. *Biological Psychiatry*, 76(4), 281–288.

6. May have missed it, but please emphasize that the observers and demonstrators were of the same sex (i.e., females with females) - if that is indeed the case.

7. Can the authors produce a reference for measuring Oxt in CSF of mice using the Enzo ELISA kit.

Reviewer #3 (Remarks to the Author):

Overall this is a nice story. The authors look at a novel test for emotional contagion in which one mouse is being shocked and the other catches the fear as evidenced by freezing. This is a rudimentary form of emotional communication as it tells us that the observer mouse recognizes that the demonstrator is in a bad way. It could be that the observer thinks that s/he is being shocked (does not recognize the distinction between self and other) or that s/he is catching the other's distress while recognizing the self - other distinction. Either way, this meets the fundamental definition of empathy as a communication of affect between animals.

This manuscript describes a set of convincing and comprehensive experiments that demonstrate a critical role for oxytocin in a rudimentary form of murine empathy. Using the freezing reaction of a mouse to a conspecific's shock-evoked distress as a model for empathy, the authors investigated the effects that nasal oxytocin administration, activation of oxytocinergic neurons, and oxytocin receptor antagonism have on empathy, demonstrating that oxytocin and oxytocin receptor-mediated pathways play important roles in empathy.

1. The current study is not the first investigation of the role of oxytocin in murine empathy. 46-47: This is not the first investigation of oxytocin in empathy in mice with oxytocin. There has of course been a great deal of work on the contribution of oxytocin to maternal care behavior, which is commonly thought to be the prototypical form of empathy. Moreover, Lukas et al. (2011, *Neuropsychopharmacology*) looked at the effects of oxytocin receptor antagonist on prosocial

behaviors in mice; Langford et al. (2009, Social Neuroscience) showed that knocking out the oxytocin receptor did not affect mice's social approach to conspecifics in pain. Both studies used manipulations of the effectiveness of oxytocin in the mouse nervous system and observed causal effects of such manipulations. And of course, there are studies looking at oxytocin and empathy or affective communication in closely related rodent species – rats and voles.

Nevertheless, the current study expands greatly on the findings of previous studies.

2. Please show control trials.

In suppl Fig 2 it would appear that the non-shocked/no demonstrator baseline averages for freezing are ~3-4 AU. Do these control conditions show the same upward trend from acclimation to the trials and then across trials? Seeing control time courses would be valuable to determine if in fact males freeze for unfamiliar albeif less than for familiar or only freeze for familiar.

The sequence of events for the two sequential controls is confusing. Both controls are listed in Fig 1a but neither is referenced in Fig 1b. Did all mice get both control conditions?

3. The Conditioned Place Avoidance protocol is difficult to interpret and does not fit in with the rest of the experiments.

In classical conditioning, there are paired presentations of an unconditioned stimulus (here, the demonstrator rat) and a conditioned stimulus (here, the chamber closer to the demonstrator rat) that lead to an association between the conditioned stimulus and conditioned response (avoidance). It is therefore critical to examine the effect of the conditioned stimulus in the absence of the unconditioned stimulus. That was not the case in the CPA paradigm used here, where the demonstrator rat is present during testing. Thus, the test used in this manuscript simply is not conditioning because the unconditioned stimulus is present during testing.

The authors' interpretation of the CPA results is based on the assumption that since shows empathy. However, whether this assumption stands is questionable. This would appear to be a stretch on a fairly small and delimited difference in behavior. For one, empathy has been associated with approach toward distressed conspecifics NOT avoidance of the same. Studies have reported that rodents tend to approach, rather than avoid, distressed individuals (e.g., Eg Langford et al (2009) in Social Neuroscience). This is also supported by studies investigating empathy-driven pro-social behaviors in rodents (e.g. Ben-Ami Bartal et al., 2011 in Science): that rodents approach distressed conspecifics and perform actions to help suggests that avoidance of noxious stimuli may not apply to empathic contexts. Furthermore, familiar individuals get helped more (see Ben-Ami Bartal et al., 2014 in eLife), suggesting that familiar individuals may get more approach, rather than less, in some contexts. The validity of the test is therefore questionable.

In the "CPA" test, animals continued to avoid the chamber containing a familiar demonstrator rat. This did not happen when the demonstrator was unfamiliar. The authors' interpret the persistent avoidance of the demonstrator rat chamber as empathy. It is unclear why this is empathy rather than fear. Previous studies have reported that rodents tend to approach, rather than avoid, distressed individuals (e.g., Eg Langford et al (2009) in Social Neuroscience). This is also supported by studies investigating empathy-driven pro-social behaviors in rodents (e.g. Ben-Ami Bartal et al., 2011 in Science): that rodents approach distressed conspecifics and perform actions to help suggests that avoidance of other animals exhibiting distress may not in fact be empathy. Furthermore, familiar individuals get helped more (see Ben-Ami Bartal et al., 2014 in eLife), suggesting that familiar individuals may get more approach, rather than less, in some contexts. The validity of the test is therefore questionable. [PM1]

Finally, even if mice find familiar conspecifics in distress [PM2] more aversive than unfamiliar conspecifics in distress, it is not clear that the mice are empathizing with the familiar. It may be that familiar elicit a stronger fear response than unfamiliar. A more granular analysis of the animals' behavior within the chamber closer to the demonstrator and in connection with the

demonstrator (orientation toward or away, for example) would add some heft to this claim.

Adding to the concern about the “CPA” study are the variable sample sizes. For example: familiar mice (n=10) and non-familiars (n=12) in the sCPA expt in Fig 2b left. Then familiar mice (n=10) and non-familiars (n=8) in Fig 2b right. So 10 and 12 for the “acclimation and conditioning” time course graph but 10 and 8 for acclimation bar graph. Obviously, this raises questions as to why some data were included for only select analyses. Without a clear justification for the variant sample sizes, the data are severely compromised.

In sum, the fact that this is not a conditioning paradigm, the sample size concerns, and the lack of oxytocin tie-in for this experiment all support omitting the “CPA” test from the manuscript.

4. Possible sensory processing differences

98-100: “Not due to deficits in the capacity ...” Certainly the orienting response means that the mice are “detecting” the sensory signals, but they could be “processing” them differently. Of course there may be differences in the demonstrators’ behaviors toward familiars and unfamiliar (other than call duration) that were not measured or recognized by the investigators. For these reasons the conclusion that “deficits in the transmission of fear from unfamiliar mice were not a result of deficits in the capacity to detect or process sensory signals emanating from these mice” should be toned down.

5. Vocalization analysis

One parameter (duration) of one type of call (alarm, ~22 kHz) was measured. That this measure did not differ across groups is certainly not proof that the animals are emitting the same collective information across the entirety of the sensory modalities.

295-7: The freezing connection to vocalizations is confusing. It appears that freezing only correlated with vocalizations in familiar animals. In unfamiliar females contagion happened but vocalization levels did not predict it. Doesn’t this suggest that vocalizations are not responsible for the communication that leads to freezing?

6. Oxytocin’s effect on social memory

A critical issue that the authors should address is whether oxytocin manipulation (adding or antagonizing) is affecting the mouse’s ability to recognize familiar conspecifics. As cited in this manuscript, Ferguson et al. (2000, Nature) showed that Oxt -/- mice have social amnesia: they don’t remember mice that they’ve met. Could it be that oxytocin is modulating familiarity, and then secondarily, having an effect on the measures that are interpreted as signs of empathy? This doesn’t necessarily counter the authors’ overall story but it may merit discussion.

7. Individual familiarity and genetic relatedness are confounded.

If understood correctly, the “familiar” mice are littermates who have always lived together. If this is the case, and if the authors have never done this experiment with cross-fostered mice, then it is unclear whether the mice are importantly “familiar” or genetically related. The authors choose to call the mice familiar and thus emphasize the experience over the genetics. Whether this is warranted cannot be assessed by the data provided. It is recommended that the term littermate be used to reduce unwarranted emphasis.

361-373: When discussing the paradoxical observations that oxytocin generally reduces anxiety in rodents and that oxytocin enhances empathically transmitted affect (distress including anxiety), the authors propose that a resolution could be that the anxiolytic effects of oxytocin are secondary to its prosocial effects. However, it is unclear how this could explain oxytocin-anxiety studies where rodents are almost always conditioned and tested individually (and therefore not in a social context). While it is true, as the authors state, that the presence of conspecifics buffers anxiety, it is unclear how oxytocin could have mimicked the effect of social buffering in studies of individual animals.

8. Relevance of the periventricular nucleus to the story.

The PVN experiments come out of the blue. Do PVN neurons project to amygdala? Are these projecting neurons OXT-containing? Is PVN the primary source of OXT to amygdala? A search of the ms revealed no text on this topic and no references with either PVN or periventricular in the title. This choice needs to be better motivated, or the motivation needs to be better explained to the reader.

9. Choice of terms

Why are the authors not using the term emotional contagion? How does emotional state-matching differ from emotional contagion? What led to this choice of terminology?

10. Connection to receptor internalization

352-4: Isn't internalization simply a result of chronic receptor activation? I don't see that that would necessarily predict a down regulation in transcript. I do see that overstimulation appears to have down regulated OXTR transcript. But I don't see the relevance of the internalization data. Indeed in ACC there is no OXTR transcript downregulation.

11. Context for chronic effect.

The persistent chronic effect, weeks after limited administration of OXT, is truly remarkable. Why did the authors think this would work? Is there some background that would help place this experiment in context? Is there relevant clinical data?

Minor comments:

41: Oxytocin as a "substrate" of empathy is awkward. It certainly "is an important" modulator in social behavior.

89. Although it is obvious to the authors and to many readers that developmental experiences alter adult social behavior and affect, adding a review reference will increase the accessibility of this ms.

In fig 3c I cannot see a difference in the shades of the i.n. acute and chronic saline. If it is there it is too subtle. If it's not there, this should be fixed. In general the color choices in this figure are not the easiest to detect (eg pink vs red may be lost on a significant portion of male readers). Instead of shades of gray, how about white and gray? Instead of two shades of gray and black in earlier figures how about white gray and black or white and black? These changes would make these figures more quickly pop out at the reader.

Fig 1d is presented as large vs small litters but the difference is between familiar and unfamiliar mice. The text would be easier to follow if the figure mirrored it.

CNO is not defined. Understood that this is the agonist for the DREADD but nonetheless, it should be so stated.

Reviewer #1:

Major points

1. *The authors do not document what the Dem are doing during the paradigm, leaving unclear what is being transmitted between animals and complicating the interpretation of these effects as 'empathy'. The authors refer to what the Dem is doing sometimes as 'displaying distress', which seems fair (and loose), but other times they say that it is showing Pavlovian fear conditioning. Yet there is no documentation of this. In fact, and maybe I am not fully understanding the paradigm, but if indeed the Dem is being fear conditioned to the context, shouldn't the animal exhibit fear responses already during the acclimation period of Cond2? If yes, shouldn't those responses also be transmitted to the Obs? In Fig. 1f, we see that the Obs is not freezing during acclimation period of Cond2 and Cond3. This either means that the Obs is not really responding to freezing per se (but we do not have any correlation of behaviour between the Dem/Observer pair besides vocalizations) or that the Dem did not learn. The only documentation shows that the observer is sensitive to the duration of vocalization (Fig. 1c) and that this measure does not change with OTX experiment – this is solid. But these experiments do not establish how much of the social effect on observer freezing is accounted for by this measure and how other modes of transmission may contribute. Therefore, I would like to have the report of the Dem's behaviour.*

We have measured demonstrator fear behaviors (freezing during conditioning; potentiated startle at 24hrs post-conditioning). These data (Sup Fig 1,4) indicate that demonstrators exhibited freezing during conditioning, and their degree of freezing correlates with that of the observer only when the two animals were familiar. Furthermore, we have run an additional experiment to assess the effects of fear conditioning in demonstrators 24hrs post-conditioning. The demonstrators exhibited contextual and cued fear-potentiated startle at this time point, indicating that they had acquired fear during conditioning and in the presence of the observer. This strengthens our contention that the increase in activity we saw on Cond2 and Cond3 reflected efforts to escape, which are incompatible with the expression of freezing. This contention was supported by our study using a new behavioral paradigm showing that the observers do indeed make efforts to escape, as shown in Sup Fig 6 and Fig2b. We acknowledge that vocalizations may not be the sole form of demonstrator-emitted cue to contribute to the transfer of fear to the observer. Hence we write in the discussion only that our findings suggest that vocalizations "contribute" to social fear.

2. *The OTX experiments have adequate but not very thorough controls. I would have liked to see the authors perform the manipulation experiments on both familiar and unfamiliar observers. Additionally, since CNO can have physiological effects, a control for CNO injection on Sham surgery animals would be helpful.*

We have completed both suggested experiments. Intranasal oxytocin administration to familiar observers does not significantly enhance freezing (Sup Fig 7), likely implying a ceiling effect. We have also added a chemogenetic control group in which Oxt-Cre transgene-expressing mice infected with a conditional GFP-encoding virus are injected with CNO prior to social fear conditioning. In this new control group we observed freezing levels comparable to the existing control group in which mice were injected with the DREADD and administered saline.

3. *The effect of litter size, which is mainly a side-point, appears to be a post-hoc analysis. I wonder if this might be a statistical artefact resulting from checking multiple possible data splits which were not reported. If the authors wish to support this result a retest would greatly enhance confidence in this result.*

Indeed, this analysis of litter size was post-hoc, although it happened that we had evenly and well-defined small (2-6 pups) versus large (7-12 pups) litter size differentiation. We have been unable to retest this phenomenon because of poorer luck in obtaining an adequate and even range of litter sizes. However, we can assert that this effect is not an artifact resulting from an arbitrary or opportune data split; the effect

holds if we expand the definition of “small” litters to <8 pups (rather than <7 pups): small (2-7 pups; Familiar n = 5, Unfamiliar n= 3) vs large (8-12 pups; Familiar n = 8, Unfamiliar n = 10) - ANOVA: $F_{3,25} = 3.84$, $p = 0.024$; effect of familiarity, $F_{1,25} = 10.5$, $p = 0.004$; effect of size $F_{1,25} = 0.078$, $p = 0.783$; familiarity x size interaction, $F_{3,25} = 2.3$, $p = 0.142$). Therefore, the effect is robust and, we believe, warrants inclusion within Figure 1.

4. The paradigm includes not only shock conditioning, which is the main focus, but also startle. This is only revealed in the Supplementary info and is not really described in the main text and methods. I found this confusing. I wonder what the effects of the various conditions were on this second paradigm. The authors do show that OTX increases startle reactivity (Fig S6a) but don't comment on it. This result seems to be in disagreement to the main result, which is increase in freezing responses. I also wonder if there are any interactions between paradigms. Overall, I see why the authors would want to simplify the story, but I think they cannot keep this can of worms partly open.

Potentiated startle is a very well established measure of fear and anxiety. It is important to note that it is not a different form of fear conditioning, merely an additional behavioral measure of fear. Since fear-potentiated startle and freezing are both established, and highly correlated, measures of fear (Leaton – J Exp Psych - 1990) the elevation in startle is in fact consistent with our primary finding that oxytocin enhances freezing. Even though we did not include measurement of startle in all experiments, we hope that the use of these two measures of fear will be seen as a strength of the paper, rather than as a potential liability.

5. SCPA experiment seems to distract from the main story. This paradigm which involves ‘escape’ is not as well documented. As it is an active response, it may have more to do with the reactivity assay than the fear assay, but this was not clearly described.

We understand that this paradigm is not well documented; however, we believe it contributes meaningfully to interpretation of our findings. This escape-related paradigm is novel, reinforces our conclusion that observers are expressing fear, and stands to expand our understanding of social fear as an adaptive and ethologically relevant process. (As we describe in the manuscript, escape behavior is the primary response of ground squirrels in a naturalistic setting to the alarm calls of a conspecific, but only if that animal is from the same burrow (Sherman – Science – 1977)). It also aids significantly in explaining the increased activity levels noted in Sup Fig 6, which was the reason we developed the paradigm in the first place.

Minor points:

1. The authors should mention two studies in rodents, Atsak et al, Plos1 2011 and Pereira et al, Current Biology 2012. The first study shows very similar results to the present study and the second one is a clear example of species difference. In the case of rats, prior experience is necessary for the exhibition of social transmission of fear.

References to these studies have been incorporated into the text (within the introduction), and we have included a statement concerning prior experience (within the discussion).

2. Fig 1e – The authors should show the baseline movement here in order to allow comparison. Otherwise it is hard to know whether there is an absolute effect across both groups at M1. This seems to be true from Fig 1f.

We added this baseline/acclimation activity comparison to Fig1e.

3. I would like to see a more thorough discussion of the main result with OXT administration. According to Gozzi et al, Neuron, OXT administration leads to decrease in freezing response due to a direct effect on CeL cells that inhibit CeM (blocking freezing). However, in this paradigm OXT leads to an increase in

freezing response on the Obs. How do the authors reconcile these two effects? Gozzi et al show the effect on a non-social paradigm. How do you think these two treatments can lead to disparate outcomes?

This is addressed in our discussion. The reason for these contrasting findings is unclear. We propose (albeit speculatively) that the disparity between findings by Gozzi and our group stem from social milieu. Oxytocin is thought to modulate “social salience” (whether of positive or negative valence), and may have altogether different effects in non-social contexts. Further, our data would suggest that oxytocin acting within the ACC may underlie this social-specific enhancement in fear. In this regard, our findings are buttressed by the recent report (Burkett – Science - 2016) that the oxytocin antagonist OXTA administered systemically or into the ACC reduced “allogrooming” (a measure of “consolation behavior”).

4. In general, I would appreciate a better presentation of the results, it is hard to compare locomotion activity with freezing counts, when those terms sometimes are almost used interchangeably.

We have done our best to illustrate (in Fig 1a) and separate data depicting freezing (Fig 1b-d) and activity measures (Fig 1e and Sup Fig 6).

Reviewer #2

The manuscript by Pisansky and colleagues investigates the role of oxytocin in a model of mouse empathy. The study quite interesting and well-performed.

There are only a few items for the authors to address:

1. One can question whether empathy is the best description for mice. Emotional contagion provides an alternative explanation for this matched state and does not require Theory of Mind, which has not been documented in mice. This is a fundamental criticism of the terms used in this manuscript and could shift the scope of this research. It should be noted though that empathy and emotional contagion are not mutually exclusive and yet distinctions should be made, when appropriate. This could be discussed.

Emotional contagion is a term more commonly used in the human literature for a phenomenon that parallels affective state matching. We agree with the reviewer that emotional contagion/affective state matching does not require a cognitive “theory of mind.” However, in referring to this phenomenon as a form of empathy we are conforming to the way in which empathy has been defined by a number of leading theorists in the fields of animal (e.g., de Waal) and human (e.g., Baron-Cohen) behavior. Indeed, several other papers have also used the term “empathy” when describing similar behaviors in rodents (see Burkett – Science - 2016; Chen - PLoS One - 2009) and one opinion article (Grenier - Nature Neuro - 2010) has concluded that: “These social modulations correspond to at least a broad behavioral definition of empathy.” We also draw attention to comments from Reviewer 3 (below): “Either way, this meets the fundamental definition of empathy as a communication of affect between animals.” Within our discussion we do discuss our behavior as “a form of empathy” and note that expression of fear contagion in humans is associated with other measures of empathy.

2. The authors note that oxytocin receptor expression in the ACC are not affected by chronic nasal administration of oxytocin and conclude a distinction in the role of oxytocin in acute vs chronic regulation of ACC activity as it related to empathy. However, it should be noted that the authors made no attempt to confirm that nasal administration of oxytocin affects local concentrations of oxytocin or oxytocin-related function in the ACC. Thus, the absence of change in the ACC could be a product of the manipulation used.

As described in our note to the editor above, we now report significant levels in the ACC as well as other brain structures after intranasal delivery of the I¹²⁵-labeled oxytocin. These findings closely mirror the distribution of other intranasally-administered neuropeptides in rats (Thorne – Neuroscience – 2004, Dhuria – J Pharm Sci – 2008), and of regional cerebral blood flow after oxytocin administration in mice, recently reported (Galbusera - Neuropsychopharmacology - 2017). As further confirmation of our findings, we have also demonstrated increased expression of c-fos after oxytocin administration in the ACC but not

in the neighboring primary motor region (M1) that does not contain oxytocinergic projections (Knobloch – Neuron - 2012).

3. *The DREADD manipulation is not very conservative, given the wide-spread of oxytocin projections in the brain that originate from the PVN, including those involved directly in fear conditioning. Could the authors discuss or specify the scale of effect that this manipulation had on brain function?*

Indeed, oxytocinergic projections extend to a number of areas of the mammalian forebrain (see Knobloch – Neuron - 2012). While we're unable to identify specific projections that underlie our chemogenetic effects, our intranasal oxytocin data suggests at least two regions – the ACC and amygdala – as promising sites in the regulation of oxytocin's effects on social fear, and therefore empathy.

4. *The authors mention a couple of times that oxytocin has no role in fear learning. However, see, at least:*

- Pagani, J. H., Lee, H.-J., & Young, W. S. (2011). Postweaning, forebrain-specific perturbation of the oxytocin system impairs fear conditioning. *Genes, Brain, and Behavior*, 10(7), 710–719.

and

- Modi, M. E., Majchrzak, M. J., Fonseca, K. R., Doran, A., Osgood, S., Vanase-Frawley, M., et al. (2016). Peripheral Administration of a Long-Acting Peptide Oxytocin Receptor Agonist Inhibits Fear-Induced Freezing. *The Journal of Pharmacology and Experimental Therapeutics*, 358(2), 164–172.

These references have been added (within the discussion). We have also stated (within the introduction) that oxytocin has known influences on learning.

5. *An important study of “empathy” in rodents not cited is:*

Smith, A. S., & Wang, Z. (2014). Hypothalamic oxytocin mediates social buffering of the stress response. *Biological Psychiatry*, 76(4), 281–288.

This reference has been added (within the introduction).

6. *May have missed it, but please emphasize that the observers and demonstrators were of the same sex (i.e., females with females) - if that is indeed the case.*

This has been added (within the introduction and methods).

7. *Can the authors produce a reference for measuring Oxt in CSF of mice using the Enzo ELISA kit.*

We have included reference to the manufacturer Enzo Life Sciences, Inc. (within the methods).

Reviewer #3:

Overall this is a nice story. The authors look at a novel test for emotional contagion in which one mouse is being shocked and the other catches the fear as evidenced by freezing. This is a rudimentary form of emotional communication as it tells us that the observer mouse recognizes that the demonstrator is in a bad way. It could be that the observer thinks that s/he is being shocked (does not recognize the distinction between self and other) or that s/he is catching the other's distress while recognizing the self - other distinction. Either way, this meets the fundamental definition of empathy as a communication of affect between animals.

This manuscript describes a set of convincing and comprehensive experiments that demonstrate a critical

role for oxytocin in a rudimentary form of murine empathy. Using the freezing reaction of a mouse to a conspecific's shock-evoked distress as a model for empathy, the authors investigated the effects that nasal oxytocin administration, activation of oxytocinergic neurons, and oxytocin receptor antagonism have on empathy, demonstrating that oxytocin and oxytocin receptor-mediated pathways play important roles in empathy.

1. The current study is not the first investigation of the role of oxytocin in murine empathy.

46-47: This is not the first investigation of oxytocin in empathy in mice with oxytocin. There has of course been a great deal of work on the contribution of oxytocin to maternal care behavior, which is commonly thought to be the prototypical form of empathy. Moreover, Lukas et al. (2011, Neuropsychopharmacology) looked at the effects of oxytocin receptor antagonist on prosocial behaviors in mice; Langford et al. (2009, Social Neuroscience) showed that knocking out the oxytocin receptor did not affect mice's social approach to conspecifics in pain. Both studies used manipulations of the effectiveness of oxytocin in the mouse nervous system and observed causal effects of such manipulations. And of course, there are studies looking at oxytocin and empathy or affective communication in closely related rodent species – rats and voles.

Nevertheless, the current study expands greatly on the findings of previous studies.

We recognize that these previous studies also looked at aspects of empathy, broadly defined. We have therefore updated the introduction to avoid the suggestion that this is the first murine model of empathy.

2. Please show control trials.

In suppl Fig 2 it would appear that the non-shocked/no demonstrator baseline averages for freezing are ~3-4 AU. Do these control conditions show the same upward trend from acclimation to the trials and then across trials? Seeing control time courses would be valuable to determine if in fact males freeze for unfamiliar albeit less than for familiars or only freeze for familiars.

These data have been added to existing graphs. While there is an upward trend in freezing behavior from acclimation to “conditioning” trials in these control experiments, the magnitude of effect is far smaller than that seen during conditioning.

The sequence of events for the two sequential controls is confusing. Both controls are listed in Fig 1a but neither is referenced in Fig 1b. Did all mice get both control conditions?

All mice received both control conditions prior to social conditioning, as stated. Data from the control experiments are found within the supplementary data section.

3. The Conditioned Place Avoidance protocol is difficult to interpret and does not fit in with the rest of the experiments.

In classical conditioning, there are paired presentations of an unconditioned stimulus (here, the demonstrator rat) and a conditioned stimulus (here, the chamber closer to the demonstrator rat) that lead to an association between the conditioned stimulus and conditioned response (avoidance). It is therefore critical to examine the effect of the conditioned stimulus in the absence of the unconditioned stimulus. That was not the case in the CPA paradigm used here, where the demonstrator rat is present during testing. Thus, the test used in this manuscript simply is not conditioning because the unconditioned stimulus is present during testing.

The purpose of our study was to examine social transmission of fear rather than vicarious, or observational, fear conditioning, although we agree that this would be an interesting future direction to pursue. In our description of the task, we do not suggest that observers acquire conditioned avoidance behavior to the chamber. To avoid the implication that the observers necessarily exhibited associative learning, we have renamed the paradigm “socially induced avoidance” (see above).

The authors' interpretation of the CPA results is based on the assumption that since shows empathy. However, whether this assumption stands is questionable. This would appear to be a stretch on a fairly small and delimited difference in behavior. For one, empathy has been associated with approach toward distressed conspecifics NOT avoidance of the same. Studies have reported that rodents tend to approach, rather than avoid, distressed individuals (e.g., Eg Langford et al (2009) in Social Neuroscience). This is also supported by studies investigating empathy-driven pro-social behaviors in rodents (e.g. Ben-Ami Bartal et al., 2011 in Science): that rodents approach distressed conspecifics and perform actions to help suggests that avoidance of noxious stimuli may not apply to empathic contexts. Furthermore, familiar individuals get helped more (see Ben-Ami Bartal et al., 2014 in eLife), suggesting that familiar individuals may get more approach, rather than less, in some contexts. The validity of the test is therefore questionable.

The phenomenon we are describing is commonly referred to as “affective state matching” or “emotional contagion.” As we wrote above, our classification of this phenomenon as a form of empathy is consistent with the way in which empathy has been construed by a variety of scientists in this field, including Baron-Cohen, de Waal, Lahvis, and Panksepp. An animal that (or, indeed, person who) performs pro-social behaviors to help a distressed individual displays “compassion” or “sympathy”, states that are, as the reviewer suggests, likely to be “empathy-driven”. But “empathy-driven”, by definition, is not the same as “empathy” itself. In other words, empathy can evoke different behaviors, depending on the context. We would contend that the fact that the terms compassion/sympathy and empathy are sometimes used interchangeably should not preclude use of the term “empathy” in the absence of prosocial behavior.

Finally, even if mice find familiar conspecifics in distress [PM2] more aversive than unfamiliar conspecifics in distress, it is not clear that the mice are empathizing with the familiars. It may be that familiars elicit a stronger fear response than unfamiliar. A more granular analysis of the animals' behavior within the chamber closer to the demonstrator and in connection with the demonstrator (orientation toward or away, for example) would add some heft to this claim.

We have incorporated measurement of demonstrator fear behavior (freezing) during conditioning (see Sup Fig 4). Further, familiar and unfamiliar demonstrators freeze equally. We note a correlation in freezing between familiar observer-demonstrator pairs, and not unfamiliar pairs (Sup Fig 1). Combined with data showing that demonstrators do not vocalize differently depending on familiarity (Sup Fig 5), we conclude that familiar demonstrators do not elicit a stronger fear response compared to unfamiliar demonstrators.

Adding to the concern about the “CPA” study are the variable sample sizes. For example: familiar mice (n=10) and non-familiars (n=12) in the sCPA expt in Fig 2b left. Then familiar mice (n=10) and non-familiars (n=8) in Fig 2b right. So 10 and 12 for the “acclimation and conditioning” time course graph but 10 and 8 for acclimation bar graph. Obviously, this raises questions as to why some data were included for only select analyses. Without a clear justification for the variant sample sizes, the data are severely compromised.

The same mice are represented throughout Fig 2. Group sizes are variable because several mice escaped the testing apparatus during at least one of experimental days. This has been mentioned in the “Statistics” section within the Methods.

In sum, the fact that this is not a conditioning paradigm, the sample size concerns, and the lack of oxytocin tie-in for this experiment all support omitting the “CPA” test from the manuscript.

This was not a conditioning paradigm per se but we realize that the name we chose (social conditioned place preference) suggested otherwise. We believe this paradigm is highly informative, for reasons described above. We have thus retained it in the manuscript but changed its name to the more precise, “socially induced avoidance”.

4. Possible sensory processing differences

98-100: *“Not due to deficits in the capacity ...” Certainly the orienting response means that the mice are “detecting” the sensory signals, but they could be “processing” them differently. Of course there may be differences in the demonstrators’ behaviors toward familiars and unfamiliar (other than call duration) that were not measured or recognized by the investigators. For these reasons the conclusion that “deficits in the transmission of fear from unfamiliar mice were not a result of deficits in the capacity to detect or process sensory signals emanating from these mice” should be toned down.*

We have updated this sentence by removing “or process”.

5. Vocalization analysis

One parameter (duration) of one type of call (alarm, ~22 kHz) was measured. That this measure did not differ across groups is certainly not proof that the animals are emitting the same collective information across the entirety of the sensory modalities.

See also response to Reviewer 1. We suggest in the manuscript that vocalizations “contribute” to social transmission of fear. We do not suggest, and nor would we wish to suggest, that this is the only relevant aspect of the demonstrator’s behavior in transmitting fear to the conspecific.

295-7: The freezing connection to vocalizations is confusing. It appears that freezing only correlated with vocalizations in familiar animals. In unfamiliar females contagion happened but vocalization levels did not predict it. Doesn’t this suggest that vocalizations are not responsible for the communication that leads to freezing?

Once again, we are not suggesting that vocalizations are the sole source of transmission of fear-related information between the two mice. In the case of the female mice however, it is also likely that a high level of freezing in female observers militated against detection of a significant correlation with demonstrator vocalizations (i.e., a ceiling effect).

6. Oxytocin’s effect on social memory

A critical issue that the authors should address is whether oxytocin manipulation (adding or antagonizing) is affecting the mouse’s ability to recognize familiar conspecifics. As cited in this manuscript, Ferguson et al. (2000, Nature) showed that Oxt -/- mice have social amnesia: they don’t remember mice that they’ve met. Could it be that oxytocin is modulating familiarity, and then secondarily, having an effect on the measures that are interpreted as signs of empathy? This doesn’t necessarily counter the authors’ overall story but it may merit discussion.

This is an interesting suggestion; however, we think it is unlikely that oxytocin is acting by modulating social memory. For one, unfamiliar pairs have no prior social experience (within the home-cage or via direct interaction within the testing chambers) for oxytocin to improve upon. Secondly, intranasal oxytocin administered to familiar observers does not enhance socially transmitted fear, or activity during acclimation that would suggest recognition of the demonstrator.

7. Individual familiarity and genetic relatedness are confounded.

If understood correctly, the “familiar” mice are littermates who have always lived together. If this is the case, and if the authors have never done this experiment with cross-fostered mice, then it is unclear whether the mice are importantly “familiar” or genetically related. The authors choose to call the mice familiar and thus emphasize the experience over the genetics. Whether this is warranted cannot be assessed by the data provided. It is recommended that the term littermate be used to reduce unwarranted emphasis.

This is a very good point, which applies to a number of studies in this area (e.g., Jeon – Nat Neuro - 2010; Ito – Neuropsychopharm - 2015; both cited in the manuscript), including our own. We now draw attention to this issue in the discussion. It will be of interest whether, having now raised the issue, future

research will disambiguate the effects of rearing environment vs genetic relatedness on empathic behavior. We do not believe, however, that interpretation of the other findings contained in the current study rests on this issue's resolution.

361-373: When discussing the paradoxical observations that oxytocin generally reduces anxiety in rodents and that oxytocin enhances empathically transmitted affect (distress including anxiety), the authors propose that a resolution could be that the anxiolytic effects of oxytocin are secondary to its prosocial effects. However, it is unclear how this could explain oxytocin-anxiety studies where rodents are almost always conditioned and tested individually (and therefore not in a social context). While it is true, as the authors state, that the presence of conspecifics buffers anxiety, it is unclear how oxytocin could have mimicked the effect of social buffering in studies of individual animals.

Rodents in classical fear conditioning experiments, while tested in isolation, are often housed with conspecifics. Hence there are likely to be uncontrolled "social buffering" effects (see, e.g., Guzman – Psychopharm - 2014). Furthermore, in studies in which oxytocin is given we suspect that it is common for the drug to be administered to more than one animal at a time such that animals receive the drug in the company of conspecifics. Since few studies explicitly control for the effects of social context on fear processes, home-cage interactions, or the presence of a conspecific during drug administration, may still contribute to the effects of oxytocin on fear conditioning.

8. Relevance of the periventricular nucleus to the story.

The PVN experiments come out of the blue. Do PVN neurons project to amygdala? Are these projecting neurons OXT-containing? Is PVN the primary source of OXT to amygdala? A search of the ms revealed no text on this topic and no references with either PVN or periventricular in the title. This choice needs to be better motivated, or the motivation needs to be better explained to the reader.

The PVN is the primary oxytocin-producing nucleus of the mammalian brain. Oxytocinergic neurons of the PVN project to several regions of the forebrain, including to the amygdala and ACC.

9. Choice of terms

Why are the authors not using the term emotional contagion? How does emotional state-matching differ from emotional contagion? What led to this choice of terminology?

These terms have been used fairly interchangeably in the empathy literature (see, e.g., Panksepp - Neurosci Biobehav Rev - 2011; de Waal - Annu Rev Psych - 2008). We prefer the former term because it is more descriptive and does not imply – as it may in some readers' minds – that this adaptive capacity is pathological. We include the latter term in both the introduction and discussion since some readers may be more familiar with it.

10. Connection to receptor internalization

352-4: Isn't internalization simply a result of chronic receptor activation? I don't see that that would necessarily predict a down regulation in transcript. I do see that overstimulation appears to have down regulated OXTR transcript. But I don't see the relevance of the internalization data. Indeed in ACC there is no OXTR transcript downregulation.

We agree. We have changed the text to state that our measurement of receptor expression downregulation may have the same functional effect as receptor internalization. While these phenomena are not the same, both are adaptive processes elicited upon extended exposure to oxytocin.

11. Context for chronic effect.

The persistent chronic effect, weeks after limited administration of OXT, is truly remarkable. Why did the authors think this would work? Is there some background that would help place this experiment in context? Is there relevant clinical data?

We cite several studies in humans (Yatawara – Molec Psych - 2015) and rodent species (Bales – Mole Psych - 2014; Huang – Neuropsychopharm - 2013) that serve as our rationale for pursuing chronic intranasal oxytocin administration. Dosing of oxytocin is an important factor in the effects of oxytocin, and these data contribute to this growing body of work.

Minor comments:

41: Oxytocin as a “substrate” of empathy is awkward. It certainly “is an important” modulator in social behavior.

We have replaced “a substrate”, with “a molecular substrate”, which we think captures its role fairly well.

89. Although it is obvious to the authors and to many readers that developmental experiences alter adult social behavior and affect, adding a review reference will increase the accessibility of this ms.

We have added a reference to this end (Meaney – Animal Behav 1981), placed within the Results.

In fig 3c I cannot see a difference in the shades of the i.n. acute and chronic saline. If it is there it is too subtle. If it's not there, this should be fixed. In general the color choices in this figure are not the easiest to detect (eg pink vs red may be lost on a significant portion of male readers). Instead of shades of gray, how about white and gray? Instead of two shades of gray and black in earlier figures how about white gray and black or white and black? These changes would make these figures more quickly pop out at the reader.

We have done our best to use the most intuitive and easily identifiable figure colors throughout.

Fig 1d is presented as large vs small litters but the difference is between familiar and unfamiliar mice. The text would be easier to follow if the figure mirrored it.

The current design of this sub-figure remains consistent with the remainder of the figure – that is, black indicating familiar mice; gray indicating unfamiliar mice.

CNO is not defined. Understood that this is the agonist for the Dredd but nonetheless, it should be so stated.

We have updated the text.

Reviewers' comments:

Reviewer #1 (Remarks to the Author):

The revised ms "Oxytocin enhances empathy in mice" by Dr Gewirtz and colleagues is greatly improved and I am fully satisfied by the way they addressed the majority of the issues raised by the reviewers, including two new experiments and extra analysis.

In particular, I really appreciate the fact that the authors performed the control CNO experiment (which is missing from many high profile papers) and the experiment with radiolabeled oxytocin. It clearly shows that indeed intranasal oxytocin administration can penetrate several brain regions in mice and be the cause for the results obtained. These two experiments give very solid support to their findings.

Several minor issues related to the presentation of the results were also addressed and in my opinion made the ms much easier to follow.

In summary, the revised ms is greatly improved and I do not have extra concerns.

Reviewer #2 (Remarks to the Author):

The authors' addressed my previous comments mostly.

1. However, despite their assertion, they did not in fact include the Pagani reference.

2. Also, the measurement of 125-I in the brain after administration of 125-I-labeled Oxt does not measure Oxt. The labeled Oxt could rapidly be metabolized and a labeled amino acid or even free 125-I could be present, without any delivered oxytocin. At a minimum, the label extracted from the brain should have been run on a chromatograph in comparison with the originally labeled Oxt. Perhaps the ELISA kit should be used to measure Oxt concentrations in the ACC after IN Oxt administration.

3. A manufacturer's catalog number of the ELISA kit should be supplied in the methods

Reviewer #3 (Remarks to the Author):

This is a largely responsive revision. While this reviewer continues to have some of the same concerns as previously as well as a few new ones (e.g. escape appears to be very loosely defined so that anything that was not freezing could be construed as escape; this is not the well defined darting of Gruene et al 2015).

1. Addition of Pagani reference

“The authors’ addressed my previous comments mostly. However, despite their assertion, they did not in fact include the Pagani reference.”

We apologize for this oversight and have now incorporated this valuable reference into the manuscript.

2. Measurement of oxytocin levels in the anterior cingulate cortex (ACC) after intranasal administration.

We thank the reviewer for these comments as well as those contained in his/her previous review. We agree that the fate of intranasally administered oxytocin in the CNS is an issue requiring careful consideration, especially given the wide use of this approach in many studies in humans and other species, and its potential therapeutic application. With the additional experiments we included in the revised manuscript, we believe we have demonstrated – to the extent possible given currently available methodologies (see below) – that intranasally administered oxytocin penetrates the CNS, including the ACC, to elicit the empathy-like behavioral effects observed in this study. Our rationale for reaching this conclusion is described below, as well as a description of relevant changes we have made to the manuscript to address the reviewer’s concerns.

“... the measurement of 125-I in the brain after administration of 125-I-labeled Oxt does not measure Oxt. The labeled Oxt could rapidly be metabolized and a labeled amino acid or even free 125-I could be present, without any delivered oxytocin.”

The balance of evidence does not favor these two possibilities.

1) The passage of peptide molecules has previously been shown by one of us (Leah Hanson and colleagues at HealthPartners Research Foundation) (*Thorne et al., 2004*) and others (e.g., *Balin et al., 1986*). In one such study, we reported significant levels in the brain of the radioactively labeled neuropeptide orexin after intranasal administration (*Dhuria et al., 2009*). Given the greater potential for decay catastrophe for larger peptides (*Catt et al., 1973*), oxytocin (9 amino acids) would be expected to undergo less fragmentation than the larger orexin molecule (33 amino acids). Moreover, the delivery of insulin-like growth factor-I (IGF-I) protein (70 amino acids) to the rat brain after intranasal administration was demonstrated by both 125 I labeling as well as activation of second messenger pathways using both western blotting and immunohistochemistry (*Thorne et al., 2004*). Larger yet, mesenchymal stem cells have been shown using direct detection to reach the brain after administration to the nose (*Danielyan et al. 2011*).

2) In the methods used here, detachment or degradation of the radiolabel (125 I) was minimized by following the lactoperoxidase procedure, a relatively mild form of radiolabeling that produces more stable products compared to other procedures (*Redman et al., 1984; Wajchenberg et al., 1978; Baumann et al., 1986*). Using this approach, 125 I-angiotensin II, for instance, exhibits high retention of radiolabeling (80%) for up to 5 months post-labeling (*Catt et al., 1973*).

3) In our study the administered radiolabeled peptide was purified and free 125 I removed prior to its administration.

4) Intranasal oxytocin significantly increases oxytocin levels in CSF, measured using ELISA in our study and others (e.g., *Lee et al., 2017*). This assay detects the intact peptide and not fragments or single amino acids.

5) The brain distribution pattern in our study is consistent with the pattern of oxytocin receptor binding (*Hammock et al., 2014*) and immunolabeling (*Mitre et al., 2016*) across numerous areas, including the amygdala, hypothalamus, and olfactory bulbs. Most notably, the region showing the highest oxytocin receptor immunolabeling is the suprachiasmatic nucleus of the hypothalamus (64.9±4.8%; per *Mitre et al., 2016*). We similarly found the highest CNS concentration of radiolabeled oxytocin (outside of the

olfactory bulbs/nuclei) in the hypothalamus (22.2uM). In fact, labeling was significantly greater in the hypothalamus than in the thalamus (two-tailed t-test; $t=3.0$, $p=0.02$), an adjacent diencephalic structure also exposed to the ventricles but, critically, containing much less receptor expression in adult mice (Mitre *et al.*, 2016). The significantly differential distribution of the label in and of itself is hard to reconcile *a posteriori* with the possibility that it primarily reflects the presence of fragments of the peptide or free I^{125} .

6) There is indirect evidence from numerous studies (see Evans *et al.*, 2013 and Veening & Olivier, 2013 for reviews), in the form of physiological and behavioral effects, that intranasally administered oxytocin (as well as other, larger peptides) indeed reaches the brain in mice. Most recently, regional cerebral blood volume measurements following intranasal oxytocin showed functional changes in activity in brain regions corresponding to those containing a high density of oxytocin receptors (Galbusera *et al.*, 2016), including those highlighted in our study.

7) Further indirect evidence from the current study is that this treatment selectively activated the ACC, as shown by elevated c-Fos immunoreactivity.

“At a minimum, the label extracted from the brain should have been run on a chromatograph in comparison with the originally labeled Oxt.”

Previous chromatography analyses of rat brain samples by our (Leah Hanson and colleagues at HealthPartners Research Foundation) group and others have documented intact iodinated peptides, including large molecular weight proteins (Hanson unpublished; Kastin *et al.*, 1999) and vasopressin (Dufes *et al.*, 2003) after intranasal drug administration. Evidence of intact iodinated vasopressin is noteworthy, given its structural similarity to oxytocin. Dhuria *et al.* 2009 demonstrated intact I^{125} -orexin reaching the brain using HPLC measurement. We did not conduct a chromatography assay in the present study because it was not possible to administer enough radiation nasally to a mouse (given volume limitations and concentration of the radiolabeled ligand) to detect the peptide in the limited amount of homogenate that one can load on a gel. This is particularly true for analysis of an individual brain region, such as the ACC.

“Perhaps the ELISA kit should be used to measure Oxt concentrations in the ACC after IN Oxt administration.”

In response to the reviewer's initial review we made several attempts at this approach. Despite using two ELISA kits (Enzo Life Sciences and Arbor Assays) we were unable to extract oxytocin and therefore detect its presence in brain tissue. Indeed, to the best of our knowledge there have been no publications that have reported successful ELISA-based extraction of oxytocin from mouse brain tissue. Importantly, however, the fact that the ELISA approach revealed the presence of oxytocin within CSF provides *prima facie* evidence that oxytocin reaches the CNS intact. In addition to highlighting this in the discussion (see below), we also now note in the Results that the ELISA protocols do not yet seem to be optimized for detecting oxytocin in brain tissue (“We were unsuccessful in extracting and quantifying oxytocin from brain tissue (in contrast to CSF) using existing ELISA assays (see Methods)”). Sharing this information may be valuable to other researchers in this field seeking to optimize the use of ELISA to detect oxytocin in mouse neural tissue or tissue of other species.

To succinctly acknowledge the reviewer's concerns regarding the radiolabeling procedure, we have added the following sentence to the discussion:

“Although the I^{125} -radiolabeling technique cannot distinguish between intact peptide and labeled fragments of the peptide, detection of intact oxytocin in CSF by ELISA argues against the latter possibility in the current study, consistent with previous chromatography analyses (see ⁶⁰ for discussion). “

In summary, we believe that the case we make above is well summarized in the response of Reviewer 1: *“In particular, I really appreciate the fact that the authors performed the control CNO experiment (which is missing from many high profile papers) and the experiment with radiolabeled oxytocin. It clearly shows that indeed intranasal oxytocin administration can penetrate several brain regions in mice and be the cause for the results obtained. These two experiments give very solid support to their findings.”*

Perhaps more importantly, we suggest that the other findings in this manuscript, and their overall significance, do not rest on whether or not oxytocin levels in the ACC alone are elevated following intranasal administration. Given the distribution of oxytocin receptors in the brain, and the involvement of oxytocinergic projections to other targets, it is highly likely that the effects of exogenous oxytocin on empathic behavior relies on its actions in a system of interconnected structures. We have now added this point more explicitly to the discussion:

“The breadth of its distribution in structures rich in oxytocin receptors and implicated in other empathy-related behavioral domains suggests that exogenously applied oxytocin affects empathic behavior via its actions on a network of neural substrates such as the hypothalamus, ACC, amygdala^{14,35,64} ().”

3. “A manufacturer’s catalog number of the ELISA kit should be supplied in the methods”

This item number has been added to the methods. We have also included the catalog number for the second ELISA kit (Arbor Assays, Inc.) used – albeit unsuccessfully – to measure oxytocin from brain tissue.

REFERENCES

- Balin et al. Avenues for entry of peripherally administered protein to the central nervous system in mouse, rat, and squirrel monkey. *J Comp Neuro*. 1986.
- Baumann & Amburn. The autodecomposition of radiolabeled human growth hormone. *J Immunoassay*. 1986.
- Catt et al. Prolonged retention of high specific activity by 125I-labeled angiotensin II – a consequence of ‘decay catastrophe’. *Biochemica et Biophysica Acta*. 1973.
- Danielyan et al. Therapeutic efficacy of intranasally delivered mesenchymal stem cells in a rat model of Parkinson Disease. *Rejuvenation Medicine*. 2011.
- Dhuria et al. Intranasal drug targeting of hypocretin-1 (orexin-A) to the central nervous system. *J Pharm Sci*. 2009.
- Dufes et al., Brain delivery of vasoactive intestinal peptide (VIP) following nasal administration to rats. *J Pharm*. 2003.
- Evans et al. Intranasal oxytocin effects on social cognition: a critique. *Brain Res*. 2013.
- Galbusera et al., Intranasal oxytocin and vasopressin modulate divergent brainwide functional substrates. *Neuropsychopharm*. 2016
- Hammock et al. Oxytocin receptor ligand binding in embryonic tissue and postnatal brain development of the C57BL/6J mouse. *Front Behav Neuro*. 2013.
- Kastin & Akerstrom. Orexin A but not orexin B rapidly enters brain from blood by simple diffusion. *J Pharm Exp Ther*. 1999.
- Lee et al., Oxytocin by intranasal and intravenous routes reaches the cerebrospinal fluid in rhesus macaques: determination using a novel oxytocin assay. *Molecular Psychiatry*. 2017.
- Redman & Tustanoff. Iodination of [Tyr8] – bradykinin – comparison of chloramine-T and lactoperoxidase techniques. *J Immunoassay*. 1984.
- Thorne et al., Delivery of insulin-like growth factor-I to the rat brain and spinal cord along olfactory and trigeminal pathways following intranasal administration. *Neuroscience*. 2004.
- Veening & Olivier. Intranasal administration of oxytocin: behavioral and clinical effects, a review. *Neuro Biobeh Rev*. 2013.
- Wajchenberg et al. Preparation of iodine-125-labeled insulin for radioimmunoassay: comparison of lactoperoxidase and chloramine-T iodination. *J Nucl Med*. 1978.